# Improving the generalizability of protein-ligand binding predictions with AI-Bind

Ayan Chatterjee[1], Robin Walters[2], Zohair Shafi [2], Omair Shafi Ahmed[2], Michael Sebek [1,3], Deisy Gysi[1,3,4], Rose Yu[5], Tina Eliassi-Rad[1,2,6,7], Albert-László Barabási[1,3,8] & Giulia Menichetti [1,3,9] ✉

Identifying novel drug-target interactions is a critical and rate-limiting step in drug discovery. While deep learning models have been proposed to accelerate the identification process, here we show that state-of-the-art models fail to generalize to novel (i.e., never-before-seen) structures. We unveil the mechanisms responsible for this shortcoming, demonstrating how models rely on shortcuts that leverage the topology of the protein-ligand bipartite network, rather than learning the node features. Here we introduce AI-Bind, a pipeline that combines network-based sampling strategies with unsupervised pre-training to improve binding predictions for novel proteins and ligands. We validate AI-Bind predictions via docking simulations and comparison with recent experimental evidence, and step up the process of interpreting machine learning prediction of protein-ligand binding by identifying potential active binding sites on the amino acid sequence. AI-Bind is a high-throughput approach to identify drug-target combinations with the potential of becoming a powerful tool in drug discovery.

The accurate prediction of binding interactions between chemicals and proteins is a critical step in drug discovery, necessary to identify new drugs and novel therapeutic targets, to reduce the failure rate in clinical trials, and to predict the safety of drugs[1]. While molecular dynamics and docking simulations[2,3] are frequently employed to identify potential protein-ligand binding, the computational complexity (namely, run-times) of the simulations and the lack of 3D protein structures significantly limit the coverage and the feasibility of large-scale testing. Therefore, machine learning (ML) and artificial intelligence (AI) based models have been proposed to circumvent the computational limitations of the existing approaches[4], leading to the development of models that rely either on deep learning architectures or chemical feature representations[5–7].

Deep learning frameworks formulate the binding prediction problem as either a binary classification task or a regression task. The successful training of a binary classifier requires positive samples, pairs of proteins and ligands that are known to bind to each other, typically extracted from protein-ligand binding databases like DrugBank[8], BindingDB[9], Tox21[10], ChEMBL[11], or Drug Target Commons (DTC)[12]. Training also requires negative samples, i.e., pairs that do not interact or only weakly interact. However, the positive and the negative annotations associated with different proteins and ligands are not evenly distributed, but some proteins and ligands have disproportionately more positive annotations than negative ones, and vice-versa, an annotation imbalance learned by the ML models, which then predict that some proteins and ligands bind disproportionately

[1]Network Science Institute, Northeastern University, Boston, MA, USA. [2]Khoury College of Computer Sciences, Northeastern University, Boston, MA, USA. [3]Department of Physics, Northeastern University, Boston, MA, USA. [4]Department of Medicine, Brigham and Women's Hospital, Harvard Medical School, Boston, MA, USA. [5]Department of Computer Science and Engineering, University of California, San Diego, CA, USA. [6]Santa Fe Institute, Santa Fe, NM, USA. [7]The Institute for Experiential AI, Northeastern University, Boston, MA, USA. [8]Department of Network and Data Science, Central European University, Budapest, Hungary. [9]Channing Division of Network Medicine, Department of Medicine, Brigham and Women's Hospital, Harvard Medical School, Boston, MA, USA. ✉e-mail: giulia.menichetti@channing.harvard.edu

more often than others. In other words, the ML models learn the binding patterns from the degree of the nodes in the protein-ligand interaction network, neglecting relevant node metadata, like the chemical structures of the ligands or the amino acid sequences of the proteins[5,13]. This annotation imbalance leads to good performance as quantified by the Area Under the Receiver Operating Characteristics (AUROC) and the Area Under the Precision Recall Curve (AUPRC) for the unknown annotations associated with missing links in the protein-ligand interaction network used for training. A key signal of such shortcut learning is the degradation of the performance of an ML model when asked to predict binding between novel (i.e., never-before-seen) protein targets and ligands. This modeling limitation is in-line with the findings of Geirhos et al.[14], who showed that deep learning methods tend to exploit shortcuts in training data to achieve good performance. Laarhoven et al. discuss similar bias in drug-target interaction data and its effect on cross-validation performance[15]. Lee et al.[16] and Wang et al.[17] proposed approaches that partly address shortcut learning, but fail to generalize to unexplored proteins, i.e., proteins that lack sufficient binding annotations, or originate from organisms with no close relatives in current protein databases. More recently, models such as MolTrans[18], MONN[19], and TransDTI[20], explore innovative structural representations of protein and ligand molecules. Though these models better leverage the molecular structures to predict binding, end-to-end training limits their ability to generalize beyond the molecular scaffolds present in the training data.

Here, we introduce AI-Bind, a pipeline for predicting protein-ligand binding which can successfully generalize to unseen proteins and ligands. AI-Bind combines network science methods with unsupervised pre-training to control for the over-fitting and the annotation imbalance of existing libraries. We leverage the notion of shortest path distance on a network to identify distant protein-ligand pairs as negative samples. Combining these network-derived negatives with experimentally validated non-binding protein-ligand pairs, we ensure sufficient positive and negative samples for each node in the training data. Additionally, AI-Bind learns, in an unsupervised fashion, the representation of the node features, i.e., the chemical structures of ligand molecules or the amino acid sequences of protein targets, helping circumvent the model's dependency on limited binding data. Instead of training the deep neural networks in an end-to-end fashion using binding data, we pre-train the embeddings for proteins and ligands using larger chemical libraries, allowing us to generalize the prediction task to chemical structures, beyond those present in the training data.

## Results

### Limitations of existing ML models

ML models characterize the likelihood of each node (proteins and ligands) to bind to other nodes according to the features and the annotations in the training data. While annotations capture known protein-ligand interactions, features refer to the chemical structures of proteins and ligands, which determine their physical and chemical properties, and are expressed as amino acid sequences or 3D structures for proteins, and chemical SMILES[21] for ligands. In an ideal scenario, the ML model learns the patterns characterizing the features which drive the protein-ligand interactions, capturing the physical and chemical properties of a protein and of a ligand that determine the mutual binding affinity. Yet, as we show next, multiple state-of-the-art deep learning models, such as DeepPurpose[5], ignore the features and rely largely on annotations, i.e., the degree information for each protein and ligand in the drug-target interaction (DTI) network, as a shortcut to make new binding predictions. A bipartite network represents the binding information as a graph with two different types of nodes: one corresponding to proteins (also called targets, representing for example, human or viral proteins) and the other corresponding to ligands (representing potential drugs or natural

compounds), respectively. A protein-ligand annotation, i.e., evidence that a ligand binds to a protein, is represented as a link between the protein and the ligand in the bipartite network[22]. Experimentally validated annotations define the known DTI network. While binding depends only on the detailed chemical characteristics of the nodes (proteins and ligands), as we show here, many ML models predictions are primarily driven by the topology of the DTI network. We begin by noticing that the number of annotations linked to a protein or a ligand follows a fat-tailed distribution[22], indicating that the vast majority of proteins and ligands have only a small number of annotations, which then coexist with a few hubs, nodes with an exceptionally large number of binding records[22]. For example, the number of annotations for proteins follows a power law distribution with degree exponent $\gamma_p = 2.84$ in the BindingDB data used for training and testing Deep-Purpose, while the ligands have a degree exponent $\gamma_l = 2.94$ (Fig. 1a). For these degree exponents, the second moment of the distribution diverges for large sample sizes, implying that the expected uncertainty in the binding information is highly significant, limiting our ability to predict the binding between a single protein and a ligand[22,23]. Furthermore, positive and negative annotations are determined by applying a threshold on kinetic constants like the constant of dissociation $K_d$. If the kinetic constant associated with a protein-ligand pair is less than a set threshold, we consider that pair as a positive or binding sample; otherwise, the pair is tagged as negative or non-binding. However, $K_d$ is not randomly distributed across the records, but the number of annotations $k$ and the average $K_d$ per $k$ (i.e., $\langle K_d \rangle$), calculated as the average across all links stemming from nodes of degree $k$, are anti-correlated (Fig. 1b), indicating stronger binding propensity for proteins and ligands with more annotations ($r_{Spearman}(k_p, \langle K_d \rangle) = -0.47$ for proteins, $r_{Spearman}(k_l, \langle K_d \rangle) = -0.29$ for ligands in the BindingDB data used by DeepPurpose). Furthermore, we observe lower variability in $K_d$ values across links originating from high-degree nodes, compared to lower-degree nodes (see Supplementary Note 1). As the annotations follow fat-tailed distributions, the observed anti-correlation drives the hub proteins and ligands to have disproportionately more binding records on average, whereas proteins and ligands with fewer annotations have both binding and non-binding examples. This annotation imbalance prompts the ML models to leverage degree information (positive and negative annotations) in making binding prediction instead of learning binding patterns from the molecular structures. We term this phenomenon as the emergence of topological shortcuts (see Supplementary Note 1).

To investigate the emergence of topological shortcuts, for each node $i$ with number of annotations $k_i$, we quantify the balance of the available training information via the degree ratio,

$$\rho_i = \frac{k_i^+}{k_i^+ + k_i^-} = \frac{k_i^+}{k_i},$$ (1)

where, $k_i^+$ is the positive degree, corresponding to the number of known binding annotations in the training data, and $k_i^-$ is the negative degree, or the number of known non-binding annotations in the training data (Fig. 2a, b). As most proteins and ligands lack either binding or non-binding annotations (Table 1), the resulting $\{\rho_i\}$ are close to 1 or 0 (See Fig. 1c above), these $\rho$ values represent the annotation imbalance in the prediction problem. As many state-of-the-art deep learning models, such as DeepPurpose[5], uniformly sample the available positive and negative annotations, they assign higher binding probability to proteins and ligands with higher $\rho$ (Fig. 2c, d). Consequently, their binding predictions are driven by topological shortcuts in the protein-ligand network, which are associated with the positive and negative annotations present in the training data rather than the structural features characterizing proteins and ligands.

The higher binding predictions in DeepPurpose for proteins with large degree ratios (Fig. 2c) prompted us to compare the performance

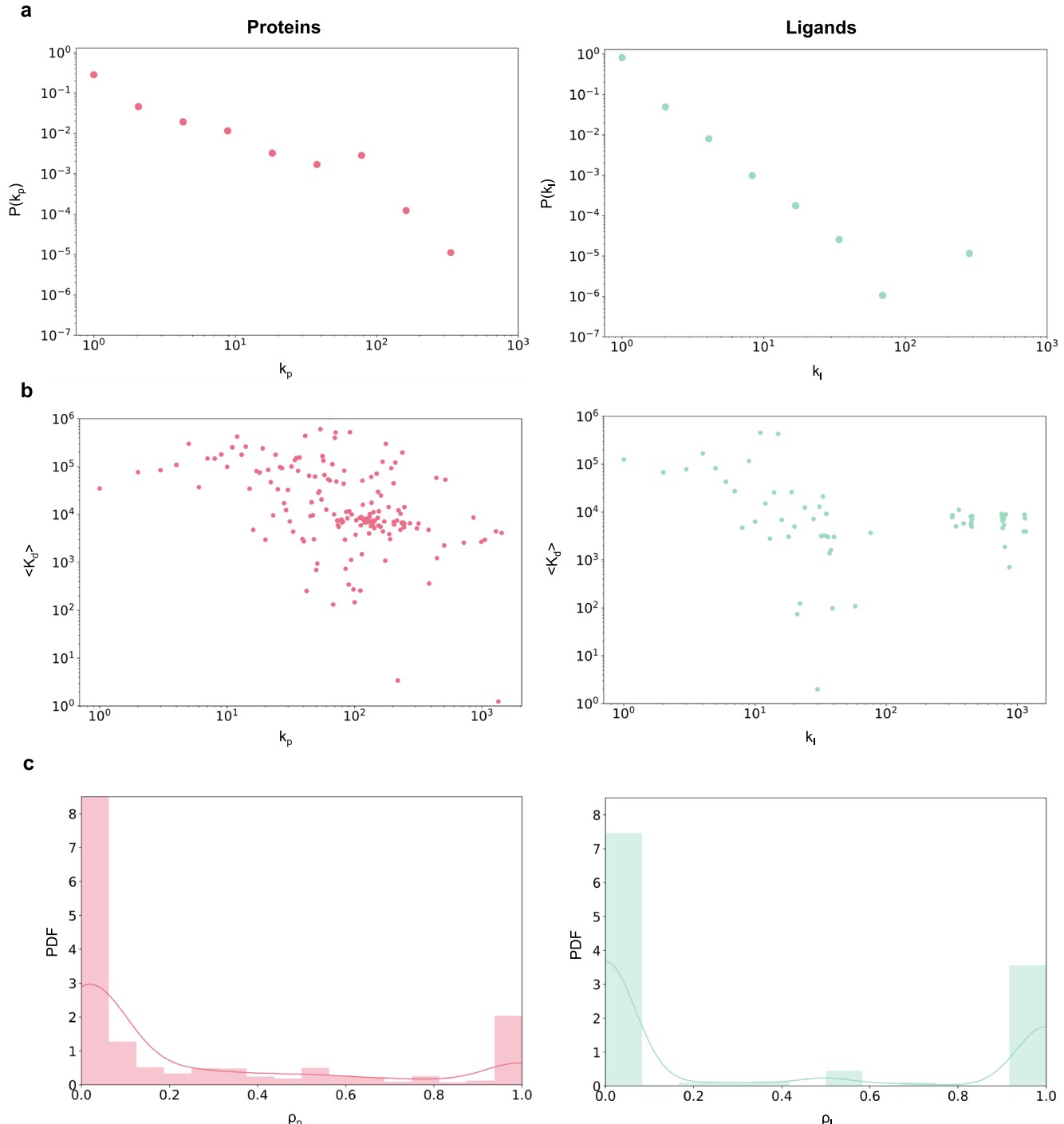

**Fig. 1 | Annotation bias in BindingDB training data and DeepPurpose predictions. a** Distributions of the number of annotations in the benchmark BindingDB data are shown in double logarithmic axes (log-log plot), indicate that $P(k_p)$ and $P(k_l)$ are well approximated by a power law for both proteins (pink) and ligands (green), with approximate degree exponents $\gamma_p = 2.84$ and $\gamma_l = 2.94$, respectively. **b** The average $K_d$ over the links for different degree values $\{k_p\}$ are negatively correlated with $r_{Spearman}(k_p, \langle K_d \rangle) = -0.47$. For the ligands, we observe similar anti-correlation with $r_{Spearman}(k_l, \langle K_d \rangle) = -0.29$. **c** The distribution of degree ratios for the proteins $\{\rho_p\}$ and the ligands $\{\rho_l\}$ in the original DeepPurpose training dataset (for a selected fold from the 5-fold cross-validation). The degree ratio, defined in

Equation (1), refers to the ratio of positive annotations to the total annotations for a given node in the protein-ligand interaction network. After thresholding $K_d$ values associated with each link to create the binary labels, the hubs on average get more positive or binding annotations, whereas the low-degree nodes get both binding and non-binding annotations. As the hubs are associated with many links in the network, learning the type of binding from the degree information helps ML models to achieve good performance by leveraging shortcut learning. The Source Data File provided with the manuscript contains the number of samples per data point in the plots.

of DeepPurpose with network configuration models, algorithms that ignore the features of proteins and ligands and instead predict the likelihood of binding by leveraging only topological constraints derived from the network degree sequence[22,24,25]. In the configuration

model (Fig. 3a, Methods), the probability of observing a link is determined only by the degrees of its end nodes. In a 5-fold cross-validation on the benchmark BindingDB dataset (Table 1), we find that the top-performing DeepPurpose architecture, Transformer-CNN[5], achieves

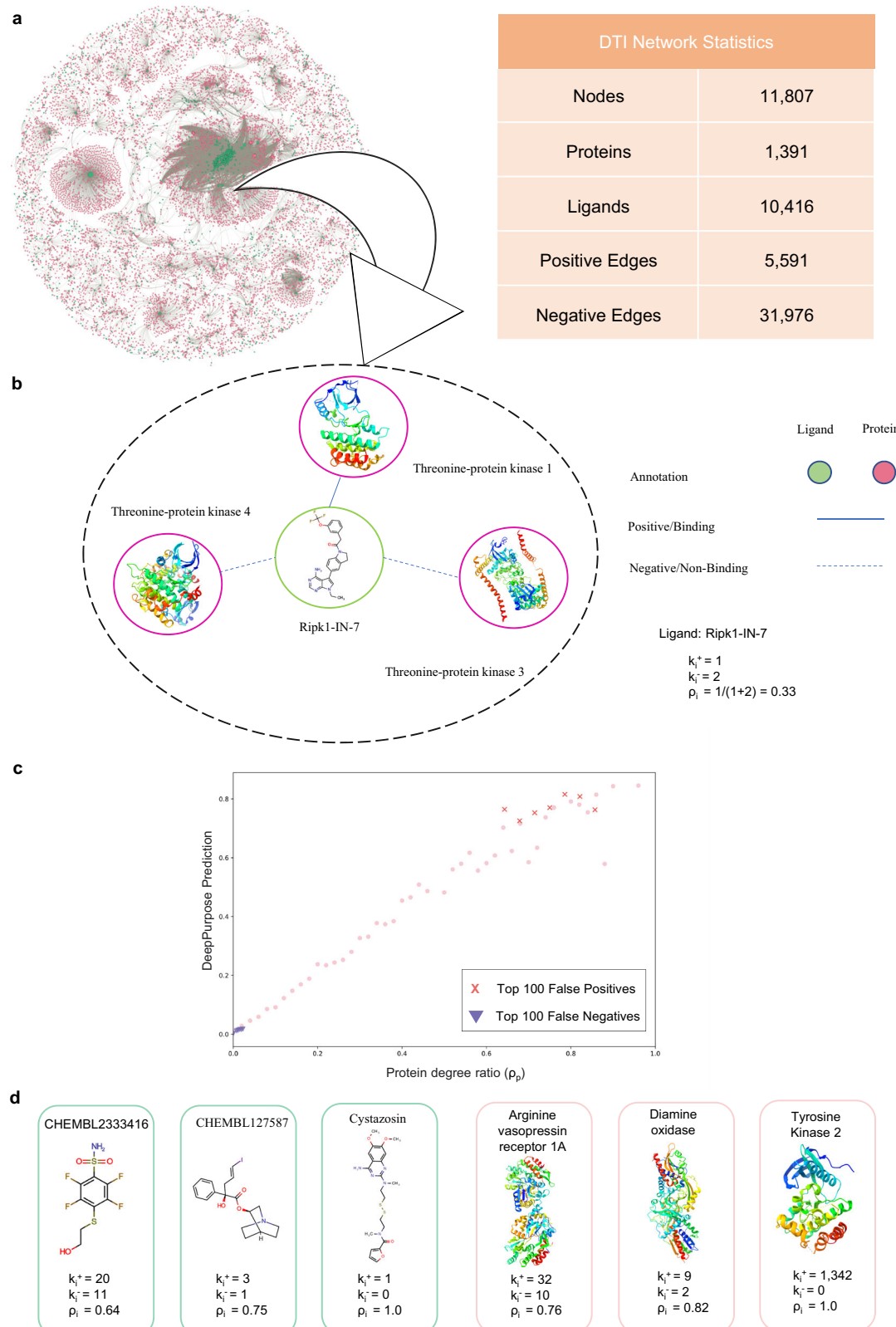

| DTI Network Statistics | |
|---|---|
| Nodes | 11,807 |
| Proteins | 1,391 |
| Ligands | 10,416 |
| Positive Edges | 5,591 |
| Negative Edges | 31,976 |

**Fig. 2 | Drug-Target Interaction Network. a** The drug-target interaction network used to train the DeepPurpose models consists of 10,416 ligands and 1391 protein targets. Ligands and proteins are represented by green and pink nodes, respectively. **b** Network neighborhood of the ligand Ripk1-IN-7. Solid links represent positive or binding annotations, while dashed links refer to negative or non-binding annotations. Ripk1-IN-7 has one positive and two negative annotations in the training data, implying a degree ratio $\rho$ of 0.33. **c** Protein degree ratios $\{\rho_p\}$ and

DeepPurpose predictions are highly correlated with $r_{Spearman} = 0.94$. We observe that the predictions for the top 100 false positive protein-ligand pairs include the proteins with large $\{\rho_p\}$ represented by the red crosses, whereas the false negative pairs are contributed by the proteins with small $\{\rho_p\}$ which are represented by the blue triangles. **d** Examples of proteins and ligands with large degree ratios, contributing to false positive predictions. Source data are provided as a Source Data file.

**Table 1 | BindingDB training data for DeepPurpose**

| Node type | Has only positive annotations | Has only negative annotations | Has both annotations | Total node count |
|---|---|---|---|---|
| Ligand | 3084 | 6539 | 793 | 10,416 |
| Protein | 168 | 556 | 667 | 1391 |

Most ligands and proteins in DeepPurpose training data have either binding or non-binding annotations, which creates imbalance in the degree ratio (see Equation (1)).

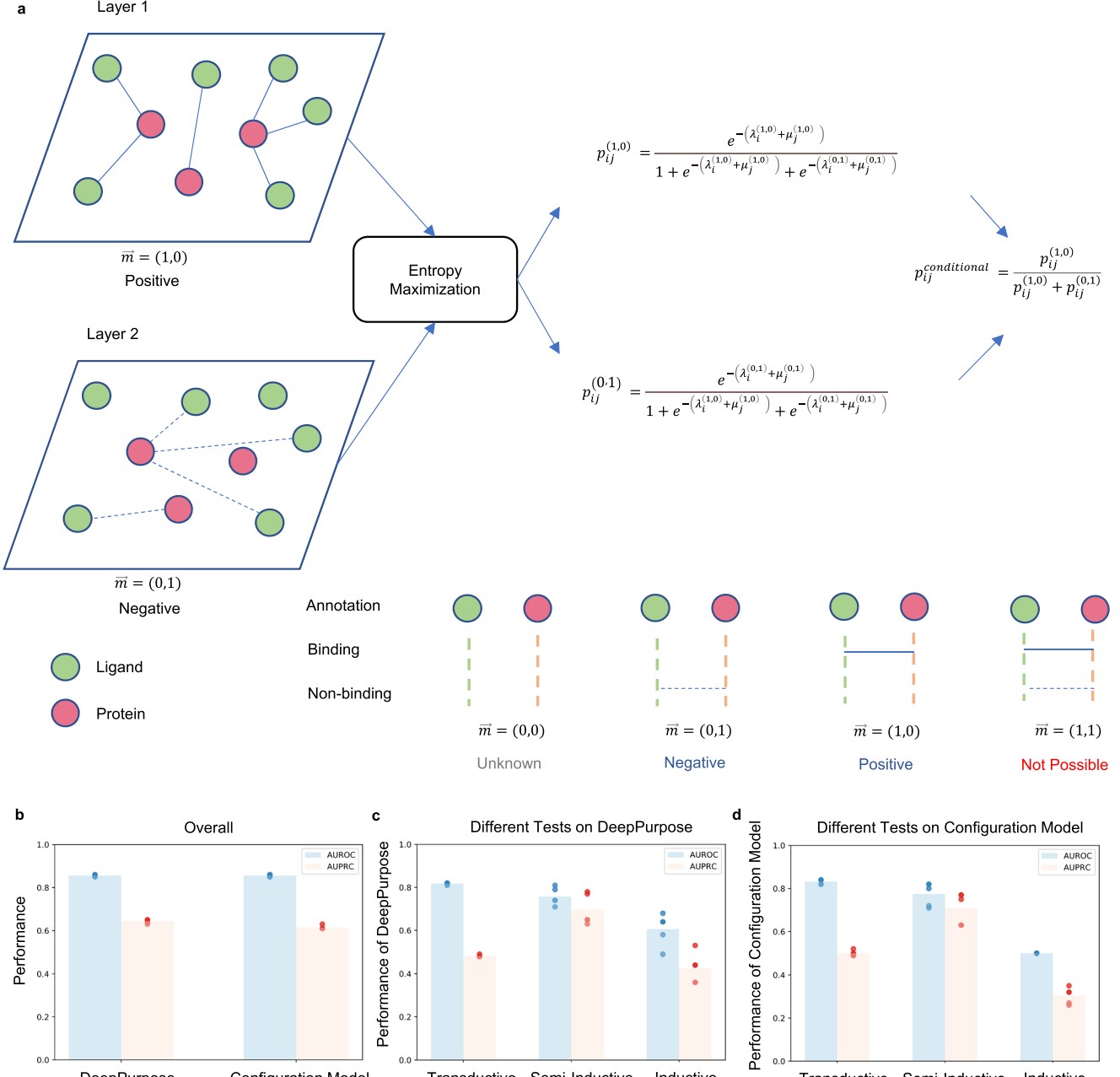

**Fig. 3 | Comparing DeepPurpose and the duplex configuration model. a** The duplex configuration model includes two layers corresponding to binding and non-binding annotations between proteins (pink nodes) and ligands (green nodes). Positive link (solid lines) and negative link (dashed lines) probabilities are determined by entropy maximization (see Methods), and used to estimate the conditional probability in transductive (Equation (7)), semi-inductive (Equation (8)), and inductive (Equation (9)) scenarios. **b**–**d** The average performance of the configuration model achieves similar results as DeepPurpose on the benchmark BindingDB data in a 5-fold cross-validation (dots represent the performance of each fold, bar height corresponds to the mean, $n = 5$). Breakdown of performances shows good predictive performance in transductive and semi-inductive scenarios. However, the same models have poor predictive performance in the inductive setting. Source data are provided as a Source Data file.

**Table 2 | DeepPurpose and duplex configuration model performances on BindingDB dataset**

| Model | Transductive | | Semi-inductive | | Inductive | |
|---|---|---|---|---|---|---|
| | AUROC | AUPRC | AUROC | AUPRC | AUROC | AUPRC |
| DeepPurpose | 0.82 ± 0.004 | 0.48 ± 0.004 | 0.76 ± 0.041 | 0.70 ± 0.073 | 0.61 ± 0.074 | 0.43 ± 0.071 |
| Config. Model | 0.83 ± 0.011 | 0.50 ± 0.012 | 0.77 ± 0.055 | 0.71 ± 0.073 | 0.50 ± 0.00 | 0.30 ± 0.038 |

DeepPurpose and the duplex configuration model perform well in both transductive and inductive tests on the benchmark BindingDB data. Both models fail to achieve good performance in the inductive test, i.e., while predicting over both unseen proteins and ligands.

**Table 3 | Assigning SMILES and amino acid sequences randomly**

| Version | AUROC | AUPRC |
|---|---|---|
| Original | 0.86 ± 0.005 | 0.64 ± 0.009 |
| Randomized | 0.84 ± 0.004 | 0.62 ± 0.004 |

A random reshuffle of SMILES and amino acid sequences does not affect the performance of DeepPurpose. This outcome suggests the limitation of DeepPurpose in learning chemical structures.

AUROC of 0.86 (±0.005) and AUPRC of 0.64 (±0.009). At the same time, the network configuration model on the same data achieves an AUROC of 0.86 (±0.005) and AUPRC of 0.61 (±0.009) (Fig. 3b).

In other words, the network configuration model, relying only on annotations, performs just as well as the deep learning model, confirming that the topology of the protein-ligand interaction network drives the prediction task. The major driving factor of the topological shortcuts is the monotone relation between $k$ and $\langle K_d \rangle$, which associates a link type with the degree of its end nodes. Moreover, in BindingDB we observe that hubs encounter less variance for $\langle K_d \rangle$ compared to the low degree nodes, making the degree of the hubs a stronger predictor of the link types. Thus, the configuration model is able to achieve good test performance in predicting the link types associated with the hubs. Since hub nodes contribute to the majority of the links in the protein-ligand bipartite network, the configuration model achieves excellent test performance by making correct predictions that mainly leverage the degree information of the hubs. To further investigate this hypothesis, we tested three distinct scenarios: (i) unseen edges (Transductive test), when both proteins and ligands from the test dataset are present in the training data; (ii) unseen targets (Semi-inductive test), when only the ligands from the test dataset are present in the training data; (iii) unseen nodes (Inductive test), when both proteins and ligands from the test dataset are absent in the training data.

We find that both DeepPurpose and the configuration model perform well in scenarios (i) and (ii) (Fig. 3c, d). However, for the inductive test scenario (iii), when confronted with new proteins and ligands, both performances drop significantly (Table 2). DeepPurpose has an AUROC of 0.61 (±0.074) and AUPRC of 0.43 (±0.071), comparable to the configuration model, for which we have AUROC of 0.50 and AUPRC of 0.30 (±0.038). To offer a final piece of evidence that DeepPurpose disregards node features, we randomly shuffled the chemical SMILES[21] and amino acid sequences in the training set, while keeping the same positive and negative annotations per node, an operation that did not change the test performance (Table 3). These tests confirm that DeepPurpose leverages network topology as a learning shortcut and fails to generalize predictions to proteins and ligands beyond the training data, indicating that we must use inductive testing to evaluate the true performance of ML models.

Beyond DeepPurpose, models such as MolTrans[18] explore different structural representations of protein and ligand molecules. We investigated transductive, semi-inductive, and inductive performances for MolTrans, a state-of-the-art protein-ligand binding prediction model which uses a combination of sub-structural pattern mining algorithm, interaction modeling module, and an augmented transformer encoder to better learn the molecular structures (see Supplementary Note 8). While the innovative representation of the molecules improves upon DeepPurpose in transductive tests (AUROC of 0.952 (±0.041), AUPRC of 0.887 (±0.087)), the same representation still relies only on the training DTI and fails to generalize to novel molecular structures, as captured by the poor performance in inductive tests (AUROC of 0.572 (±0.104), AUPRC of 0.432 (±0.105)).

## AI-Bind and statistics across models

AI-Bind is a deep learning pipeline that combines network-derived learning strategies with unsupervised pre-trained node features to optimize the exploration of the binding properties of novel proteins and ligands. Our pipeline is compatible with various neural architectures, three of which we propose here: VecNet, Siamese model, and VAENet. AI-Bind uses two inputs (Fig. 4a): For ligands, it takes as input isomeric SMILES, which capture the structures of ligand molecules. AI-Bind considers a search-space consisting of all the drug molecules available in DrugBank and the naturally occurring compounds in the Natural Compounds in Food Database (NCFD) (see Supplementary Note 4), and can be extended by leveraging larger chemical libraries like PubChem[26]. For proteins, AI-Bind uses as input the amino acid sequences retrieved from the protein databases Protein Data Bank (PDB)[27], the Universal Protein knowledgebase (UniProt)[28], and GeneCards[29].

AI-Bind benefits from several novel features compared to the state-of-the-art: (a) It relies on network-derived negatives to balance the number of positive and negative samples for each protein and ligand. To be specific, it uses protein-ligand pairs with shortest path distance ≥7 as negative samples, ensuring that the neural networks observe both binding and non-binding examples for each protein and ligand (see Fig. 5, Methods, Supplementary Note 5). (b) During unsupervised pre-training, AI-Bind uses the node embeddings trained on larger collections of chemical and protein structures, compared to the set with known binding annotations, allowing AI-Bind to learn a wider variety of structural patterns. Indeed, while models like DeepPurpose were trained on 862,337 ligands and 7504 proteins provided in BindingDB, or 7307 ligands and 4762 proteins provided in DrugBank, the unsupervised representation in AI-Bind's VecNet is trained on 19.9 million compounds from ZINC[30] and ChEMBL[11] databases, and on 546,790 proteins from Swiss-Prot[31].

We begin the model's validation by systematically comparing the performance of AI-Bind to DeepPurpose and the configuration model on a 5-fold cross-validation using the network-derived dataset for transductive, semi-inductive, and inductive tests. AI-Bind's VecNet model uses pre-trained `mol2vec`[32] and `protvec`[33] embeddings combined with a simple multi-layer perceptron to learn protein-ligand binding (Fig. 4b, see Methods). We observe that the configuration model performs poorly in inductive testing (AUROC 0.5, AUPRC 0.464 ± 0.017). Due to the network-derived negatives that remove the annotation imbalance, DeepPurpose shows improved performance for novel proteins and ligands (AUROC 0.646 ± 0.023, AUPRC 0.576 ± 0.009). The best performance on unseen nodes is observed for AI-Bind's VecNet, with AUROC of 0.75 ± 0.032 and AUPRC of 0.718 ± 0.029 (see Fig. 4c and see Supplementary Table 3 for a summary of the performances). The unsupervised pre-training for ligand embeddings allows us to generalize AI-Bind to

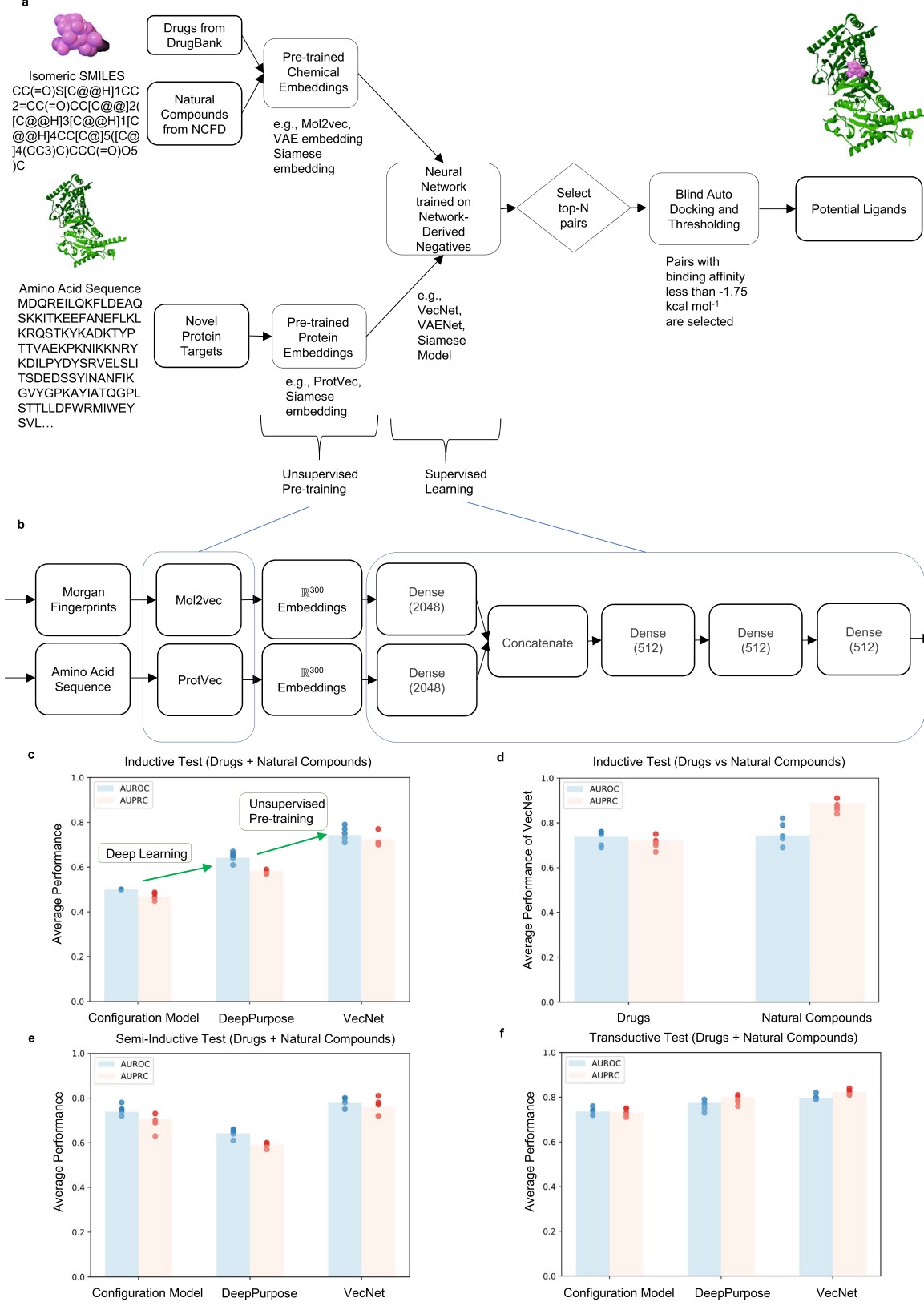

naturally occurring compounds, characterized by complex chemical structures and fewer training annotations compared to drugs (see Supplementary Note 2), obtaining performances comparable to those obtained for drugs (Fig. 4d).

Beyond DeepPurpose, AI-Bind's VecNet consistently achieves better inductive performance (AUROC $0.75 \pm 0.032$, and AUPRC $0.718 \pm 0.029$) compared to MolTrans (AUROC $0.612 \pm 0.028$, and AUPRC $0.478 \pm 0.034$). The comparison between AI-Bind and state-of-the-art models like DeepPrupose and MolTrans validates how unsupervised pre-training of the molecular embeddings improves the generalizability of binding prediction models (see Supplementary Note 8).

**Fig. 4 | AI-Bind pipeline: VecNet Performance and Validation. a** AI-Bind pipeline generates embeddings for ligands (drugs and natural compounds) and proteins using unsupervised pre-training. These embeddings are used to train the deep models. Top predictions are validated using docking simulations and are used as potential binders to test experimentally. **b** AI-Bind's VecNet architecture uses Mol2vec and ProtVec for generating the node embeddings. VecNet is trained in a 5-fold cross-validation set-up. Averaged prediction over the 5 folds is used as the final output of VecNet. **c**–**f** The average performance for a 5-fold cross-validation of VecNet, DeepPurpose, and Configuration Model (dots represent the performance of each fold, bar height corresponds to the mean, $n = 5$). All the models perform similarly in case of predicting binding for unseen edges (transductive) and unseen targets (semi-inductive). The advantage of using deep learning and unsupervised pre-training is observed in the case of unseen nodes (inductive test). AI-Bind's VecNet is the best performing model across all the scenarios. Additionally, we observe a similar performance of VecNet for both drugs and natural compounds. Source data are provided as a Source Data file.

## Validation of AI-Bind predictions on COVID-19 proteins

For a better understanding of the reliability of the AI-Bind predictions, we move beyond standard ML cross-validation and compare our predictions with molecular docking simulations, and in vitro and clinical results on protein-ligand binding. Docking simulations offer a reliable but computationally intensive method to predict (or validate) binding between proteins and ligands[34]. Motivated by the need to model rapid response to sudden health crises, we chose as our validation set the 26 SARS-CoV-2 viral proteins and the 332 human proteins targeted by the SARS-CoV-2 viral proteins[35–37]. These proteins are missing from the training data of AI-Bind, hence represent novel targets and allow us to rely on recent efforts to understand the biology of COVID-19 to validate the AI-Bind predictions. We retrieved the amino acid sequences in FASTA format for 16 SARS-CoV-2 viral proteins and 330 human proteins from UniProt[28], and use them as input to AI-Bind's VecNet. Binding between viral and human proteins is necessary for the virus to synthesize its own viral proteins and to facilitate its replication. Our goal is to predict drugs in DrugBank or naturally occurring compounds that can bind to any of the 16 SARS-CoV-2 or 330 human proteins associated with COVID-19, potentially disrupting the viral infection. After sorting all protein-ligand pairs based on their binding probability predicted by AI-Bind's VecNet ($p_{ij}^{VecNet}$), we tested the predicted top 100 and bottom 100 binding interactions with blind docking simulations using AutoDock Vina[34], which estimates binding affinity by considering all possible binding locations on the 3D protein structures (see Methods). Of the 54 proteins present in the top 100 and bottom 100 predicted pairs, 23 had 3D structures available in PDB[27] and UniProt[28], and 51 of the 59 involved ligand structures were available on PubChem[26], allowing us to perform 128 docking simulations (84 involving the top and 44 involving the bottom predictions). We find that 74 out of 84 top predictions from AI-Bind are indeed validated binding pairs. Furthermore, we find that the median binding affinity for the top VecNet predictions is −7.65 kcal mol⁻¹, while for the bottom ones is −3.0 kcal mol⁻¹ (Fig. 6a), confirming that for AI-Bind, the top predictions show significantly higher binding propensity than the bottom ones (Kruskal–Wallis $H$-test $p$-value of $2.5*10^{-5}$). As a second test, we obtained the binary labels (binding or non-binding) from docking and AI-Bind predictions using the threshold of −1.75 kcal mol⁻¹ for binding affinities[38] and the optimal threshold on $p_{ij}^{VecNet}$ corresponding to the highest F1-Score on the inductive test set (see Supplementary Note 7, Supplementary Fig. 11). In the derived confusion matrix we observe sensitivity = 0.76, representing the fraction of binding predictions made by AI-Bind that are true binders, i.e., the ratio *True Positives*/(*True Positives + False Negatives*), and F1-Score = 0.82. These two numbers confirm that the rank list provided by AI-Bind predictions shows a significant similarity to the rank list obtained by binding affinities compared to a random selection (Fig. 6b).

We further check the stability of these performance metrics by randomly choosing 20 protein-ligand pairs in a 5-fold bootstrapping set-up and observe F1-Score = $0.90 \pm 0.02$. Additionally, we find that the predictions made by AI-Bind's VecNet ($p_{ij}^{VecNet}$) and the free energy of protein-ligand binding obtained from docking ($\Delta G$) are anti-correlated with $r_{Spearman}(p_{ij}^{VecNet}, \Delta G) = -0.51$. As lower binding affinity values correspond to stronger binding, these results document the agreement between AI-Bind predictions and docking simulations.

Among the 50 ligands with the highest average binding probability we find two FDA-approved drugs Anidulafungin (NDA#021948) and Cyclosporine (ANDA#065017). Experimental evidence[39] shows that these drugs have anti-viral activity at very low concentrations in the dose-response curves, and have $IC_{50}$ values of 4.64 μM and 5.82 μM, respectively, measured by immunofluorescence analysis with an antibody specific for the viral N protein of SARS-CoV-2. These low $IC_{50}$ values support anti-viral activity, confirming that Anidulafungin and Cyclosporine bind to COVID-19 related proteins[40], and the activity at low concentrations indicate that they are safe to use for treating COVID-19 patients[1]. Anidulafungin binds to the SARS-CoV-2 viral Non-structural protein 12 (Nsp12), a key therapeutic target for coronaviruses[41].

AI-Bind also offers several novel predictions with potential therapeutic relevance. For example, it predicts that the naturally occurring compounds Spironolactone, Oleanolic acid, and Echinocystic acid are potential ligands for COVID-19 proteins, all three ligands binding to Tripartite motif-containing protein 59 (Trim59), a human protein to which the SARS-CoV-2 viral proteins Open reading frames 3a (Orf3a) and Non-structural protein 9 (Nsp9) bind[42]. Auto-Dock Vina supports these predictions, offering binding affinities −7.1 kcal mol⁻¹, −8.0 kcal mol⁻¹, and −7.6 kcal mol⁻¹, respectively.

Spironolactone, found in rainbow trout[43], has been suggested to reduce COVID susceptibility[44,45]. Oleanolic acid is present in apple, tomato, strawberry, and peach, and has been proposed as a potential anti-viral agent for COVID-19[46]. Oleanolic acid, which passed the drug efficacy benchmark ADME (Absorption, Distribution, Metabolism, and Excretion), plays an important role in controlling viral replication of SARS-CoV-2[47] and is effective in preventing virus entry at low viral loads[46]. Finally, Echinocystic acid, found in sunflower, basil, and gala apples, is known for its anti-inflammatory[48] and anti-viral activity[49], but its potential anti-viral role in COVID-19 is yet to be validated.

## Identifying active binding sites

Beyond predicting binding probability, AI-Bind can also be used to identify the probable active binding sites on the amino acid sequence, even in absence of a 3D protein structure. Specifically, we can use AI-Bind to identify which amino acid trigrams in the amino acid sequence play the most significant role in binding predictions, indicative of potential protein-ligand binding locations. We perturb each amino acid trigram in the sequence and observe the changes in AI-Bind prediction (see Supplementary Note 9). Valleys in the obtained binding probability profile represent the trigrams most predictive of binding locations on the amino acid sequence. To validate the AI-Bind predicted binding sites, we focus on the human protein Trim59, a protein for which we have results from multiple docking simulations. We visualized the binding pockets on Trim59 using PyMOL[50] and identified the amino acid residues binding to the ligand molecules (Fig. 6c). We find that the amino acid residues responsible for binding directly map to the valleys in the binding probability profile identified by AI-Bind. By viewing the docking results for Pipecuronium, Buprenorphine and Voclosporin, ligands that bind to three different pockets on Trim59, we mark the valleys corresponding to the respective binding sites on the binding probability profiles (Fig. 6c). For example, pocket 1, where Pipecuronium binds, corresponds to five AI-Bind predicted valleys marked by 1A, 1B, 1C, 1D and 1E.

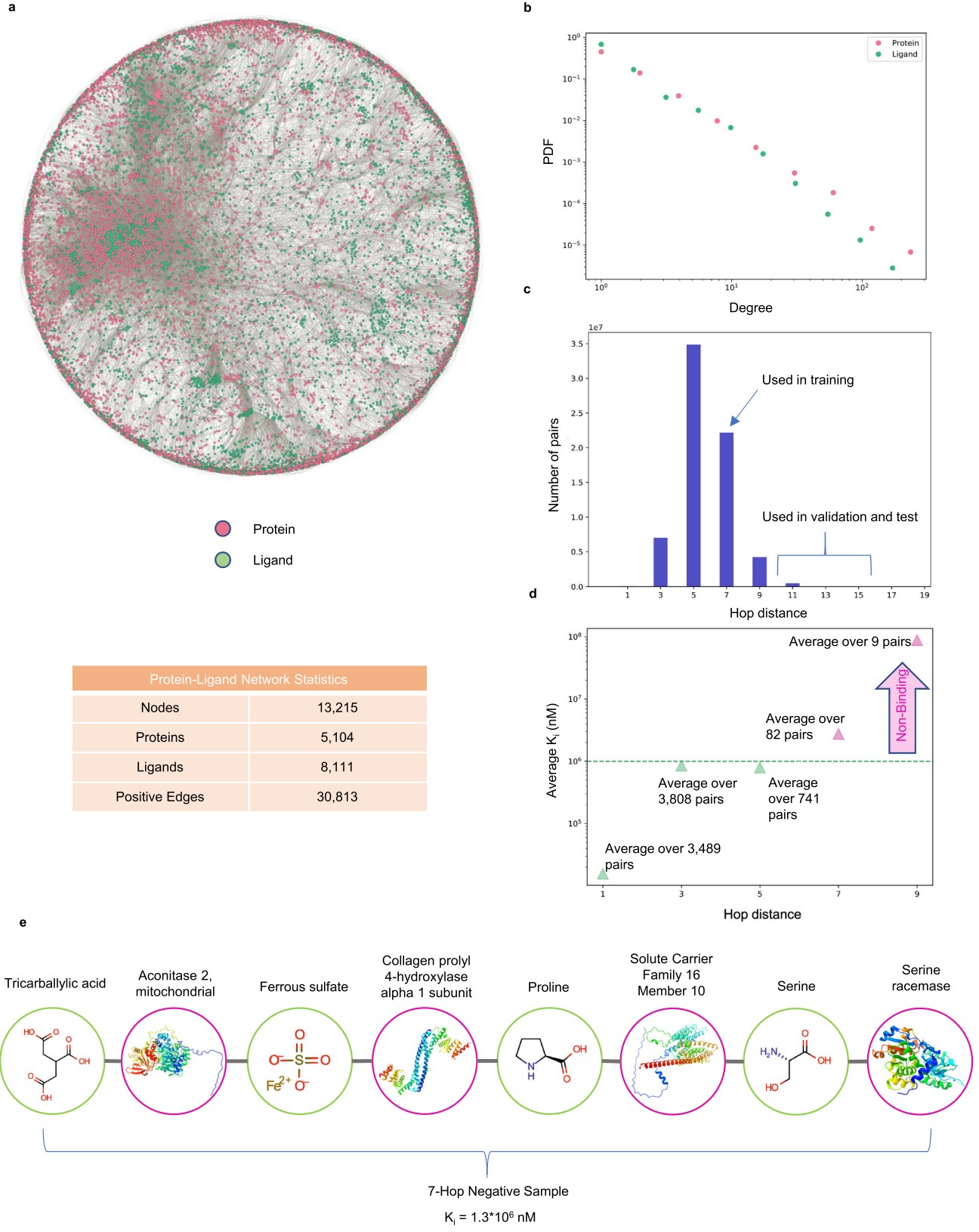

**Fig. 5 | Network-Derived Negatives. a** Protein-ligand bipartite network consisting of only binding (positive) annotations for drugs and natural compounds (green) to proteins (pink). **b** Degree distributions of ligands and proteins are fat-tailed in nature. **c** Shortest path length distribution capturing all possible protein-ligand pairs. We use protein-ligand pairs with shortest path distance of 7 for training, while absolute negatives obtained from BindingDB and pairs with shortest path distances ≥11 are used for validation and test. **d** Average experimental kinetic constant as a function of the shortest path distance. Higher path distance corresponds to higher $K_i$ in BindingDB. Beyond 7 hops, the expected constant exceeds the binding threshold of $10^6$ nM (dashed line). **e** An example of a protein-ligand pair that is 7 hops apart and is used as a negative sample in the AI-Bind training set. Source data are provided as a Source Data file.

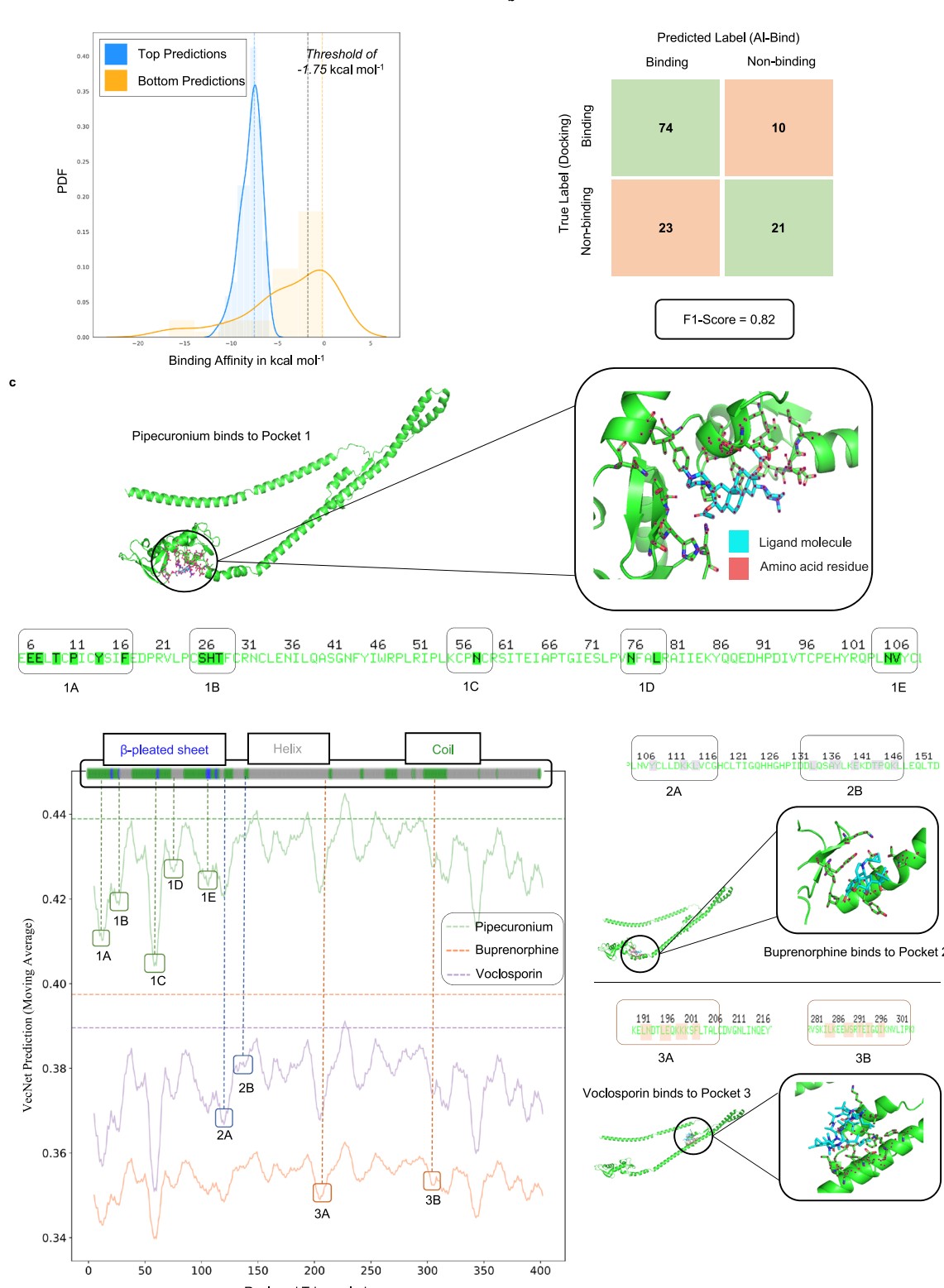

Since not all the valleys in the binding probability profile map to binding sites, we use the protein secondary structure to prioritize the valleys. We predict the secondary structure from the amino acid sequence using S4PRED[51] and identify the regions with α-helix, β-sheet and coil. In particular, α-helices prefer non-solvent accessible environments[52], contain non-polar amino acid residues[53], and consist of weaker inter-molecular interactions[54]. Thus, the presence of α-helices reduce the chances of binding between a ligand and a protein. In contrast, β-sheets and non-regular coil regions (unstructured regions) are preferred by ligands as active binding sites since they provide more binding opportunity to other molecules[55]. Indeed, most of the ligand-binding valleys in Fig. 6c map to β-sheets and coils on Trim59, associated with pockets 1 and 2 (27 out of 34 ligands validated by docking). By combining the binding probability profile predicted by AI-Bind and the secondary structure predicted by S4PRED, we can create an

**Fig. 6 | Validating and interpreting AI-Bind predictions. a** Distribution of binding affinities for top and bottom 100 predictions made by AI-Bind's VecNet over viral and human proteins associated with COVID-19. We ran docking on top 84 predictions and bottom 44 predictions. We observe that the top binding predictions (blue) of AI-Bind show lower binding energies (better binding) compared to the bottom predictions (orange). Considering the binding threshold of −1.75 kcal mol⁻¹, 88% of the top predicted pairs by AI-Bind are inline with the docking simulations. **b** We construct the confusion matrix for the top and the bottom predictions from AI-Bind. We obtain the true labels using the threshold of −1.75 kcal mol⁻¹ (gray dashed line) on the binding affinities from docking. We observe that AI-Bind predictions produce excellent F1-Score, offering predictions significantly better than random selection. **c** Binding probability profile for the human protein Trim59. Multiple valleys in the profile directly map to the amino acid residues to which the

ligands bind and are indicative of the active binding sites on the amino acid sequence. We identify the valleys on the binding probability profiles for three ligands Pipecuronium, Buprenorphine and Voclosporin, which bind at different pockets on Trim59. Valleys for these pockets have been mapped back to the amino acid sequence (valleys 1A, 1B, 1C, 1D, and 1E for pocket 1, valleys 2A and 2B for pocket 2, and valleys 3A and 3B for pocket 3). Furthermore, we highlight the secondary structure of Trim59 obtained from the amino acid sequence. Valleys containing the $\beta$-pleated sheets and the coils are more prone to binding compared to the ones with the $\alpha$-helices[52–55]. Combining the binding probability profile and the secondary protein structure allows us to identify active binding sites, guiding the design of an optimal search grid for docking simulations. Source data are provided as a Source Data file.

optimal search grid for the subsequent docking simulations, drastically reducing runtime.

We pursued further validation of AI-Bind predicted binding sites with a gold standard protein binding dataset[56] and with P2Rank, another state-of-the-art binding site prediction model[57], to extensively assess the reliability of the AI-Bind pipeline (see Supplementary Note 13).

In summary, ML models often fail in real world settings when making predictions on data that they were not explicitly trained upon, despite achieving good test performance based on traditional ML metrics. It is, therefore, necessary to validate the applicability of these models before deploying them. The documented validation of the AI-Bind predictions with molecular docking simulations and in vitro experiments offers us confidence that AI-Bind is an effective prioritization tool in diverse settings.

## Discussion

The accurate prediction of drug-target interactions is an essential precondition of drug discovery. Here we showed that by taking topological shortcuts, existing deep learning models significantly limit their predictive power. Indeed, a mechanistic and quantitative understanding of the origins of these shortcuts indicates that uniform sampling in the presence of annotation imbalance drives ML models to disregard the features of proteins and ligands, limiting their ability to generalize to novel protein targets and ligand structures. To address these shortcomings, we introduced a pipeline, AI-Bind, which mitigates the annotation imbalance of the training data by introducing network-derived negative annotations inferred via shortest path distance, and improves the transferability of the ML models to novel protein and ligand structures by unsupervised pre-training. The proposed unsupervised pre-training of node features also influences the quality of false predictions, removing potential structural biases towards specific protein families (see Supplementary Note 10). Once we improved the statistical sampling of the training data and generated the node embeddings in an unsupervised fashion, we observed an increase in performance compared to DeepPurpose, resulting in commendable AUROC (24% improvement) and AUPRC (74% improvement) and, most importantly, an ability to predict beyond proteins and ligands present in the training dataset.

A major limitation of using binding predictions in drug discovery is that binding to disease-related protein targets does not always imply a therapeutic treatment. As a future work, we plan to extend our implementation by introducing an ML-based classifier to sort the list of potential ligands according to their pharmaceutical (therapeutic) effects, combining the current node features with additional metrics derived from traditional network medicine approaches[58].

AI-Bind leverages ligands' Morgan fingerprints and proteins' amino acid sequences, which encode relevant properties of the molecules: from the presence of hydrogen donors, hydrogen acceptors, count of different atoms, chirality, and solubility for ligands, to the existence of R groups, N or C terminus in proteins. All

these properties influence the mechanisms driving protein-ligand binding (see Supplementary Note 11)[59]. Yet, the binding phenomenon is largely dependent on the 3D structures of the molecules, which determines the binding pocket structures and the rotation of the bonds. We plan to embed the 3D structures of protein and ligand molecules, which will take into account higher order molecular properties driving protein-ligand binding and refine the predictive power of AI-Bind. To maximize generalization across 3D structure, we will use SE(3) equivariant networks to learn embeddings. Equivariance has proven to be a powerful tool for improving generalization over molecular structures[60,61]. We also plan to explore the performance of AI-Bind over the entire druggable genome[62], allowing us to predict for each protein, which domains are responsible for the binding predictions. Finally, we envision enabling AI-Bind to predict the kinetic constants $K_d$, $K_i$, $IC_{50}$, and $EC_{50}$ by formulating a regression task over these variables.

The existing docking infrastructures allow screening for a specific protein structure against wide chemical libraries. Indeed, VirtualFlow[63], an open-source drug discovery platform offers virtual screening over more than 1.4 billion commercially available ligands. However, running docking simulations over these vast libraries incurs high costs for data preparation and computation time and are often limited to only proteins with 3D structures[27]. For example, in our validation step, only half (23 out of 54) of the 3D structures of the proteins associated with COVID-19 were available. Since AI-Bind only requires the chemical SMILES for ligands[21] and amino acid sequences for proteins, it can offer fast screening for large libraries of targets and molecules without requiring 3D structures, guiding the computationally expensive docking simulations on selected protein-ligand pairs.

## Methods
### Data preparation
We use InChIKeys and amino acid sequences as the unique identifiers for ligands and targets, respectively. Positive and negative samples are selected from DrugBank, BindingDB and DTC (see Supplementary Note 4). We consider samples from BindingDB and DTC to be binding or non-binding based on the kinetic constants $K_i$, $K_d$, $IC_{50}$, and $EC_{50}$. We use thresholds of $\leq 10^3$ nM and $\geq 10^6$ nM to obtain positive and (absolute) negative annotations, respectively[38]. We then filter out all samples outside the temperature range 20–45 °C to remove ambiguous pairs. All amino acid sequences were obtained from UniProt[28].

**Positive samples.** We consider the binding information from Drug-Bank as positive samples. From these annotations, we removed 53 pairs that are available in BindingDB and have kinetic constants $\geq 10^6$ nM. To obtain additional positive samples for drugs, we searched in BindingDB using their InChIKeys. We obtained 4330 binding annotations from BindingDB related to the drugs in DrugBank. Overall, we gathered a total of 28,188 positive samples for drugs. We identified also naturally occurring/food-borne compounds, small molecules

generally lacking target annotations, by leveraging the Natural Compounds in Food Database (NCFD) (see Supplementary Note 4)[64–66]. We queried BindingDB and DTC with the associated InChIKeys, obtaining a total of 1555 positive samples.

**Network-derived negative samples.** To generate annotation-balanced training data for AI-Bind, we merged the positive annotations derived from DrugBank, BindingDB, and DTC, for a total of 5104 targets and 8111 ligands, of which 485 are naturally occurring, and calculated the shortest path distribution. All odd-path lengths in the bipartite network correspond to protein-ligand pairs (Fig. 5c). Overall, the longer the shortest path distance separating a protein and a ligand, the higher the kinetic constant observed in BindingDB (Fig. 5d). In particular, pairs more than 7 hops apart have, on average, kinetic constants $K_i \geq 10^6$ nM, which is generally considered above the protein-ligand binding threshold[38] (see Supplementary Note 5). We randomly selected a subset of protein-ligand pairs which are 7 hops apart as negative samples, to create an overall class balance between positive and negative samples in the training data. Finally, we removed all nodes with only positive or only negative samples and obtained the network-derived negative instances.

We performed testing and validation on ≥11-hop distant pairs. Additionally, we included in testing and validation the absolute non-binding pairs derived from BindingDB by thresholding the kinetic constants ($K_i$, $K_d$, $IC_{50}$, and $EC_{50}$).

## Network configuration model

**Overview.** Protein-ligand annotations are naturally embedded in a bipartite duplex network, consisting of a set of nodes, comprising all proteins and ligands, interacting in two layers, each reflecting a distinct type of interaction linking the same pair of nodes[24]. More specifically, one layer (Layer 1) captures the positive or binding annotations, while the second layer (Layer 2) collects the negative or non-binding annotations (Fig. 3a). A multilink **m** between two nodes encodes the pattern of links connecting these nodes in different layers. In particular, **m** = (1, 0) indicates positive interactions, **m** = (0, 1) refers to negative interactions, **m** = (0, 0) represents the absence of any type of annotations, and **m** = (1, 1) is mathematically forbidden, as binding and non-binding cannot coexist for the same pair of protein and ligand.

We developed a canonical bipartite duplex null model that conserves on average the number of positive and negative annotations of each node, while correctly rewiring positive and negative links and avoiding forbidden configurations. By means of entropy maximization with constraints, we derive the analytical formulation of each multilink probability and the conditional probability of observing positive binding once an annotation is reported.

**Mathematical formulation.** Let $A_{ij}^{\mathbf{m}}$ be the multi-adjacency matrix representing the bipartite duplex of ligands ({$i$}) and proteins ({$j$}), with elements equal to 1 if there is a multilink **m** between $i$ and $j$ and zero otherwise. We define the multidegree of ligand $i$ and target $j$ as

$$k_i^{\mathbf{m}} = \sum_{j=1}^{N_T} A_{ij}^{\mathbf{m}}, \quad t_j^{\mathbf{m}} = \sum_{i=1}^{N_L} A_{ij}^{\mathbf{m}}, \tag{2}$$

where $N_T$ is the number of targets and $N_L$ is the number of ligands.

A bipartite duplex network ensemble can be defined as the set of all duplexes satisfying a given set of constraints, such as the expected multidegree sequences defined in Equation (2). We determine the probability of observing a bipartite duplex network $P(\vec{G})$ by entropy maximization with multidegree constraints {$k_i^{(1,0)}$}, {$k_i^{(0,1)}$}, {$t_j^{(1,0)}$}, and {$t_j^{(0,1)}$}, and corresponding Lagrangian multipliers {$\lambda_i^{(1,0)}$}, {$\lambda_i^{(0,1)}$},

{$\mu_j^{(1,0)}$}, and {$\mu_j^{(0,1)}$}[24,25]. The probability $P(\vec{G})$ factorizes as

$$P(\vec{G}) = \frac{1}{Z} \prod_{ij} \exp\left[-\sum_{\mathbf{m}\neq(0,0),(1,1)} (\lambda_i^{\mathbf{m}} + \mu_j^{\mathbf{m}}) A_{ij}^{\mathbf{m}}\right], \tag{3}$$

with

$$Z = \prod_{ij}\left[1 + \sum_{\mathbf{m}\neq(0,0),(1,1)} e^{-(\lambda_i^{\mathbf{m}} + \mu_j^{\mathbf{m}})}\right]. \tag{4}$$

Multilink probabilities $p_{ij}^{\mathbf{m}}$ are determined by the derivatives of log ($Z$) according to ($\lambda_i^{\mathbf{m}} + \mu_j^{\mathbf{m}}$). For instance, the probability of observing a positive annotation is

$$p_{ij}^{(1,0)} = \frac{e^{-(\lambda_i^{(1,0)} + \mu_j^{(1,0)})}}{1 + e^{-(\lambda_i^{(1,0)} + \mu_j^{(1,0)})} + e^{-(\lambda_i^{(0,1)} + \mu_j^{(0,1)})}}, \tag{5}$$

while the probability of observing a negative annotation follows

$$p_{ij}^{(0,1)} = \frac{e^{-(\lambda_i^{(0,1)} + \mu_j^{(0,1)})}}{1 + e^{-(\lambda_i^{(1,0)} + \mu_j^{(1,0)})} + e^{-(\lambda_i^{(0,1)} + \mu_j^{(0,1)})}}, \tag{6}$$

with $p_{ij}^{(1,0)} + p_{ij}^{(0,1)} + p_{ij}^{(0,0)} = 1$.

In this theoretical framework, binding prediction is inherently conditional, as for each ligand $i$ and protein $j$, we test only the presence of positive and negative annotations. Consequently, $p_{ij}^{(1,0)}$ and $p_{ij}^{(0,1)}$ are normalized by the probability of observing a generic annotation $p_{ij}^{(1,0)} + p_{ij}^{(0,1)}$. In case of unseen edges, binding prediction is determined by

$$p_{ij}^{\text{conditional}} = \frac{p_{ij}^{(1,0)}}{p_{ij}^{(1,0)} + p_{ij}^{(0,1)}}, \tag{7}$$

while in case of unseen target $j^*$, the binding probability towards a known compound $i$ follows

$$p_{ij^*}^{\text{conditional}} = \frac{\left\langle p_{ij}^{(1,0)}\right\rangle_j}{\left\langle p_{ij}^{(1,0)}\right\rangle_j + \left\langle p_{ij}^{(0,1)}\right\rangle_j} = \rho_i, \tag{8}$$

where $\langle\cdot\rangle_j$ denotes the average over all known targets, and $\rho_i$ follows from Equation (1). In case of unseen ligand $i^*$ and target $j^*$, the binding probability is determined by the overall number of positive ($L^{(1,0)}$) and negative ($L^{(0,1)}$) annotations, i.e.,

$$p_{i^*j^*}^{\text{conditional}} = \frac{\left\langle p_{ij}^{(1,0)}\right\rangle_{ij}}{\left\langle p_{ij}^{(1,0)}\right\rangle_{ij} + \left\langle p_{ij}^{(0,1)}\right\rangle_{ij}} = \frac{L^{(1,0)}}{L^{(1,0)} + L^{(0,1)}}, \tag{9}$$

where $\langle\cdot\rangle_{ij}$ indicates the average over all known pairs of ligands and targets.

## Novel deep learning architectures

**VecNet.** VecNet uses the pre-trained `mol2vec`[32] and `protvec`[33] models (Fig. 4b). These models create 300- and 100-dimensional embeddings for ligands and proteins, respectively. Based on `word2vec`[67], these methods treat the Morgan fingerprint[68] and the amino acid sequences as sentences, where words are fingerprint fragments or amino acid trigrams. The training is unsupervised and independent from the following binding prediction task.

**VAENet.** VAENet uses a Variational Auto-Encoder[69], an unsupervised learning technique, to embed ligands onto a latent space. The Morgan fingerprint is directly fed to convolutional layers. The auto-encoder

creates latent space embeddings by minimizing the loss of information while reconstructing the molecule from the latent representation. We train the Variational Auto-Encoder on 9.5 million chemicals from ZINC database[30], and all drugs and natural compounds in our binding dataset. Similar to VecNet, we use ProtVec for target embeddings.

**Siamese model.** The Siamese model embeds ligands and proteins into the same space using a one-shot learning approach[70]. We construct triplets of the form ⟨protein target, non−binding ligand, binding ligand⟩ and train the model to find an embedding space that maximizes the Euclidean distances between non-binding pairs, while minimizing it for the binding ones.

**File preparation for docking simulations.** We performed docking simulations for 128 protein-ligand interactions found within the top 100 and bottom 100 predictions of AI-Bind. The PDB accession codes for the 3D structures of the proteins are listed in Supplementary Table 8. The steps to implement docking simulations in AutoDock Vina[34] include:

1. Obtain the 3D ligand structures in SDF format from PubChem and save it in .pdb format with PyMOL for use in AutoDockTools.
2. Download the 3D protein structures in .pdb format and load them into AutoDockTools to remove water molecules from the protein structure, add all hydrogen atoms, and the Kollman charge to the protein.
3. Save both the protein and the ligand structures in .pdbqt format using AutoDockTools.
4. Create the grid for docking that encompasses the whole protein structure. This grid selection ensures a blind docking set-up, so that all locations on the protein are considered for determining the binding affinities. The selected grid sizes are available in gridsizes.txt (see Data availability).
5. Create the configuration files with the grid details for each protein and launch the docking simulation. We consider the protein molecules to be rigid, whereas the ligand molecules are flexible, i.e., we allow rotatable bonds for the ligands.

## Reporting summary
Further information on research design is available in the Nature Portfolio Reporting Summary linked to this article.

## Data availability
The data generated and analyzed in the study have been deposited on Zenodo at https://zenodo.org/record/7226641. The top 100 and bottom 100 binding predictions from AI-Bind on the COVID-19 related proteins are available within the Supplementary Files. A Source Data File is provided with this manuscript. The publicly available datasets used in this study can be found on their associated websites: DrugBank (https://www.drugbank.com/), BindingDB (https://www.bindingdb.org), Drug Target Commons (http://drugtargetcommons.fimm.fi/), Uniprot (https://www.uniprot.org), Protein Data Bank (https://www.rcsb.org/), and PubChem (https://pubchem.ncbi.nlm.nih.gov/).

## Code availability
The codes that support the findings of this study are openly available on our GitHub at https://doi.org/10.5281/zenodo.7730755.

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

## Acknowledgements

We thank Noah DeMoes from Lincoln Laboratory for helping out with setting up the DeepPurpose models. Christian De Frondeville from the Bioinformatics department at Northeastern University has helped with the gold standard validation of binding probability profile. This work was partially supported by NIH grant 1P01HL132825 (A.L.B.), American Heart Association grant 151708 (A.L.B.), ERC grant 810115-DYNASET (A.L.B.), and Rockefeller Foundation grant 2019 FOD 026 (A.L.B.).

## Author contributions

A.C. contributed to writing the manuscript, data curation and preparation, generating the predictions for the network configuration model, performing experiments to identify the emergence of topological shortcuts, implementing negative sample generation, developing and testing of VecNet and VAENet, running docking simulations and developing the method to predict the active binding sites. R.W. contributed to writing the manuscript, generating the predictions for the network configuration model, designing and training VecNet and VAENet. Z.S. contributed to training and testing of all the deep learning models, and designing the Siamese model. O.S.A. contributed to the deep learning literature review, running the DeepPurpose models, developing and implementing negative sample generation, training VAENet, setting up model training and software architecture as well as data pipelines for all models and experiments. M.S. contributed to exploring the optimal representation of molecules and developing the method to predict the active binding sites. D.G. contributed to the data curation and preparation, and performed the gene phylogeny study. R.Y., T.E.R., and A.L.B. have provided guidance on designing the experiments and writing the manuscript. G.M. conceived the project, developed the duplex configuration model, designed experiments to identify the emergence of topological shortcuts, contributed to data preparation, data analysis, and writing the manuscript.

## Competing interests

A.L.B. is a scientific founder of Scipher Medicine, Inc., which applies network medicine strategies to personalized drug selection, and Naring, Inc., which applies data science to food and health. The remaining authors declare no competing interests.
