## [Peer Review File · Nature Communications]

REVIEWER COMMENTS

Reviewer #1 (Remarks to the Author):

The manuscript "AI-Bind: Improving Binding Predictions for Novel Protein Targets and Ligands" describes a methodology to predict drug-target interaction. The approach covers three main steps. At first a network science based analysis is used to reduce data imbalance. Putative inactive compounds are added to the training set by selecting ligands with a shortest path ≥ 7 . Here, the distribution of data in the training set is thoroughly characterized and the performance of the AI engine is compared to a baseline model. In the next step, AI-predictions are moved into a docking workflow based on AutoDock. Perturbations are used to derive binding probability profiles.

The output of the AI-bind approach is illustrated for Covid proteins.

The authors address an important scientific problem in the field of drug discovery, the identification of the biological activity profile. The approach combines state-of-the art approaches and adds innovative concepts into the discovery approach (for example the addition of inactive data derived from the network analysis).

Many results are clearly described and presented data support the selected strategy for data augmentation.

A key validation strategy of the overall approach is cascading AI-predictions into a docking workflow and to compare docking predictions with AI predictions. Here, a true positive is given as a compound which can be docked with a good score. While this approach is frequently used and makes sense in the absence of large experimental validation data, it has its limitations. Those limitations are not adequately discussed in the manuscript, questions like is the putative binding site proven, what protein structure is used in the docking approach, how does protein flexibility and quality of the protein models influence predictions.

The link to experimental data is pointing to a cellular assay (ref. 48), while binding to specific proteins is analyzed. Functional roles of putative binding sites are not addressed.

The statement, that β -sheets are preferred as binding sites by ligands is quite bold, compared to the evidence cited. Identification of putative binding sites at the protein requires more thought. How do site detection algorithms perform in comparison to the described approach?

Results for TRIM59 do not specify what is the experimental relevance of the protein structures used. Why would binding at the three pockets influence function of the protein?

Binding probability profiles have very similar shapes. Please discuss the effect of the ligand on the binding site more clearly. Can the method help picking a putative binding site?

Minor comments:

Annotation of figure 2 should be improved (e.g. meaning of red crosses etc.)

How is the stereochemistry handled when coming from a 2D representation of the AI engine to the 3D step?

Since docking of macrocycles is challenging due to the conformational restrictions, I suggest giving more details of the procedure in the SI.

Overall, the manuscript contains some interesting approaches and studies. The largely in-silico based validation for shown ligands still casts several questions to be clarified and improved upon.

Reviewer #2 (Remarks to the Author):

AI-Bind: improving binding predictions for novel protein targets and ligands

Chatterjee et al. provided a drug-target prediction model called AI-Bind to generalize for unseen cases. Data imbalance in the training data was solved by supplementing network-derived negative samples. The features for the drugs and proteins were pre-trained from common chemical libraries.

Major comments:

1. How accurate of the network-derived negative samples are? Many drug-target databases contain only the results of positive samples, i.e. if a drug-target pair is recorded then it is a positive sample, but if a drug-target pair is not recorded then we don't know if it is truly negative, as it may be so that this drug-target pair is never tested.
2. Data that is used for pre-training of AI-Bind is unclear. The source of compound information is lacking, also, why there are more than half a million proteins? Whether the same kind of annotation imbalance on the compounds and targets exist? For example, there exist major chemical structures or protein families that are enriched in the pre-training data that in the end also drive the prediction?
3. Anidulafungin and cyclosporine have IC₅₀ of 4.64 μ M and 5.82 μ M which are not usually considered as very low concentrations. Therefore, the significance of the experimental validation is limited.
4. Only the DeepPurpose was selected for comparison. More systematic comparisons involving other state-of-the-art methods are lacking.

Reviewer #3 (Remarks to the Author):

In this work, the authors proposed a pipeline, AI-Bind, for predicting protein-ligand interactions with the generalization prediction power of unseen proteins and ligands. The network-based negative sampling, and in-depth analysis of the work could be highlighted. However, some issues must be addressed.

[Major comments]

* In Introduction, many recent DTI prediction models are neglected in this study. Please cite recent SOTA deep-learning-based models that predict DTIs or DTAs. e.g., MolTrans [1], MONN [2], KGE_NFM [3], TransDTI [4], HoTS [5], DTIHNC [6], etc.

[1] Huang, K., Xiao, C., Glass, L. M. & Sun, J. MolTrans: Molecular interaction transformer for drug target interaction prediction. *Bioinformatics* btaa880 (2020) doi:10.1093/bioinformatics/btaa880.

[2] Li, S. et al. MONN: A Multi-objective Neural Network for Predicting Compound-Protein Interactions and Affinities. *Cell Syst* 10, 308-322.e11 (2020).

[3] Ye, Q. et al. A unified drug–target interaction prediction framework based on knowledge graph and recommendation system. *Nat Commun* 12, 6775 (2021).

[4] Kalakoti, Y., Yadav, S. & Sundar, D. TransDTI: Transformer-Based Language Models for Estimating DTIs and Building a Drug Recommendation Workflow. *Acs Omega* 7, 2706–2717 (2022).

[5] Lee, I. & Nam, H. Sequence-based prediction of protein binding regions and drug–target interactions. *J Cheminformatics* 14, 5 (2022).

[6] Jiang, L. et al. Identifying drug–target interactions via heterogeneous graph attention networks combined with cross-modal similarities. *Brief Bioinform* 23, (2022).

Also, on page 3, the authors stated that the SOTA deep learning model ignores the characteristics of proteins and compounds and relies on DTI annotations. However, among recent SOTA studies, there are approaches (e.g., MolTrans, MONN, and HoTS) that learn general features of drugs and proteins in highly sophisticated ways without simply being biased toward protein-ligand and network properties. The expression of the limitations of previous studies should be rephrased.

* The authors mainly reported network characteristics using DTI information collected from BindingDB (Table 1), the same reproduction experimental conditions as the DeepPurpose model (Figure 1). The

authors argued that the DTI network has a fat-tailed distribution and said this is a common feature of the DTI network. However, since BindingDB itself is a databank for a collection of experimental assays that particular labs or researchers are interested in, it would not be surprising to observe a small number of hub nodes (proteins or ligands with high connectivity) in this DB.

If the authors want to report the generalized DTI network properties, the reviewer suggests collecting more high confident DTI data from additional DBs (e.g., DrugBank, KEGG, IUPHAR, etc.) and re-analyze the network connectivity properties.

* Performance comparisons:

- Performance comparisons with other SOTA studies that learn the general features of drugs and proteins should be addressed. Especially, as the authors highlight the generalization power for unseen proteins and unseen ligands in this study, the prediction performance for unseen proteins and unseen ligands should be compared with the aforementioned SOTAs.
- Also, in Figure 4, besides the 5-fold cross-validation performance, please provide the prediction performance using unseen split data so that readers can confirm the generalized performance.
- Please compare the results with randomly selected negative samples as well to show the effectiveness of the network-based negative sampling.
- Please provide the result of an ablation study (e.g., w/o network sampling, w/o unsupervised pre-training, etc.) so that the impact of each component of the pipeline can be distinguished.

* AutoDock simulation:

- In-depth analysis with docking simulation is interesting and could show the robustness of the method. When performing the AutoDock docking simulation, how did the authors determine the initial search box of each target protein for compounds? Need to clarify the steps in the manuscript.
- Please provide a statistical difference between the two distributions (e.g., p-value) in Figure 6A.

[Minor comments]

* On page 5, the authors argue that the topology of the protein-ligand interaction network drives the DTI prediction as some deep learning models and a network configuration model demonstrate similar prediction results. Here, the similar prediction performance does not necessarily mean that the models mainly rely on similar features. The expression should be revised.

Reviewer #1 (Remarks to the Author):

The manuscript "AI-Bind: Improving Binding Predictions for Novel Protein Targets and Ligands" describes a methodology to predict drug-target interaction. The approach covers three main steps. At first a network science based analysis is used to reduce data imbalance. Putative inactive compounds are added to the training set by selectin ligands with a shortest path ≥ 7 . Here, the distribution of data in the training set is thoroughly characterized and the performance of the AI engine is compared to a baseline model. In the next step, AI-predictions are moved into a docking workflow based on AutoDock. Perturbations are used to derive binding probability profiles.

The output of the AI-bind approach is illustrated for Covid proteins.

The authors address an important scientific problem in the field of drug discovery, the identification of the biological activity profile. The approach combines state-of-the art approaches and adds innovative concepts into the discovery approach (for example the addition of inactive data derived from the network analysis).

Many results are clearly described and presented data support the selected strategy for data augmentation.

We thank the Reviewer for the constructive feedback, and for suggesting further experiments to validate the practical usability of AI-Bind in drug discovery. We have revised our manuscript to fully address the Reviewer's comments and suggestions. In particular, we have included an additional validation of the binding probability profiles using experimentally confirmed gold standard high confidence protein-ligand binding data, and compared the AI-Bind predicted binding sites with another state-of-the-art binding site prediction technique.

1) A key validation strategy of the overall approach is cascading AI-predictions into a docking workflow and to compare docking predictions with AI predictions. Here, a true positive is given as a compound which can be docked with a good score. While this approach is frequently used and makes sense in the absence of large experimental validation data, it has its limitations. Those limitations are not adequately discussed in the manuscript, questions like is the putative binding site proven,

We fully agree with the Reviewer on the limitations of docking predictions. Prompted by the Reviewer's suggestion, we have now validated our methodology with high-confidence gold standard experimental protein-ligand binding data provided by Cheng et al [1] (see Section 'Identifying active binding sites', Lines 328-330 in the main, and SI Section 13). We show that the experimentally obtained binding locations map to the valleys of the binding probability profile. We first obtained the binding probability profile generated by AI-Bind for two different protein-ligand pairs available in the gold-standard experimental data. The gold-standard data contains binding information related to *E. Coli* protein Thymidylate Synthase, and ligands SP-722 and SP-876. We retrieved the experimentally validated secondary structure from the RCSB website, and extracted the primary binding sites of the ligand molecules. These binding locations (amino acid residues) are represented by red dots on the binding probability profiles (Figure S19). In both cases, the binding sites lie in the valleys of the binding probability profile generated by AI-Bind. Furthermore, we observe that these locations either fall on beta-sheets or coil regions. This observation is in line with our hypothesis that alpha helices have a lower

propensity for binding to the ligands. We have similar observations for the human proteins TAO3 Kinase and Human Alcohol Dehydrogenase. Additionally, we compared the AI-Bind predicted binding sites with p2rank, another state-of-the-art site prediction algorithm [2]. We observe that the AI-Bind predicted binding sites cover 64.05% of all pockets on the 195 different proteins mentioned in the gold standard dataset [1], whereas p2rank can identify 53.57% of these pockets.

2) what protein structure is used in the docking approach,

3D protein structures obtained from the PDB database or UniProt (both experimental and AlphaFold predicted) are used during validation. All protein structures are considered rigid. We consider rotatable bonds for the ligand molecules (flexible structure). Prompted by the Referee's question, we have now added a detailed description of the steps followed in the docking simulations to the manuscript under the Methods section, Lines 477-491, which includes extensive details regarding the used protein structures.

3) how does protein flexibility and quality of the protein models influence predictions?

We thank the Reviewer for raising this important question. AI-Bind predictions are not changed by flexibility and quality of the proteins as AI-Bind uses only the amino acid sequence of the proteins and SMILES of ligands for making predictions. The flexibility and quality of the proteins do impact the docking simulations. Indeed, flexible docking methods can enhance binding predictions up to 80–95% [3] for targeted docking. Indeed, by making certain regions on the protein more accessible to the ligands, it increases the chances of finding the appropriate protein conformation. Yet, the inclusion of flexible structures in blind docking increases the possibility of false positives, which are already known to be an issue [4]. Since blind docking is used to verify the AI-Bind predictions, we consider the protein structures as rigid to reduce the potential for false positives.

4) The link to experimental data is pointing to a cellular assay (ref. 48), while binding to specific proteins is analyzed.

We used ref. [62] (ref. [48] in the original manuscript) to show that two FDA-approved drugs predicted by the AI-Bind pipeline are potential antiviral drug candidates, having therapeutic effects against SARS-CoV-2 infection. We have now included ref. [55] which shows that Cyclosporine binds to SARS-CoV-2 viral proteins, supporting the prediction made by AI-Bind.

5) Functional roles of putative binding sites are not addressed.

AI-Bind in conjunction with existing experimental data can address if binding changes the functional role of a putative binding site. If the binding sites have known ligands, then it is possible to infer the changes in the functional role from existing data. If not, then experimental tests are required to fully understand how the binding site influences the function of a protein.

6) The statement, that b-sheets are preferred as binding sites by ligands is quite bold, compared to the evidence cited.

We thank the Reviewer for prompting us to clarify this statement. We have now revised Section ‘Identifying active binding sites’ of the manuscript accordingly, updating Lines 317-320. We observe that for TRIM59 there are multiple binding sites on the beta-sheets and some around the unstructured coil regions. It is possible that the affinities around the beta-sheets and the coil regions are highly protein-dependent. We now mention in the manuscript that α -helices prefer non-solvent accessible environments [5], contain non-polar amino acid residues [6], and consist of weaker intermolecular interactions [7]. Thus, we hypothesize that the regions on the binding probability profile overlapping with the alpha-helices reduce binding propensity.

7) Identification of putative binding sites at the protein requires more thought. How do site detection algorithms perform in comparison to the described approach?

This is a great suggestion. We have now added SI Section 13 where we compare AI-Bind predicted binding sites with p2rank, another machine learning-based binding site prediction method [2], for *E. Coli* and human proteins from the gold standard experimental data [1]. We compare AI-Bind and p2rank predicted binding sites to the binding sites in the gold standard dataset [1]. We observe that the AI-Bind predicted binding sites cover 64.05% of all pockets on the 195 different proteins in the gold standard dataset, whereas p2rank can identify 53.57% of these pockets. Thanks to the Referee’s recommendation, we now present three different forms of validation (comparison with docking, comparison with gold standard data, and comparison with p2rank) regarding the use of the binding probability profile for locating potential binding sites.

8) Results for TRIM59 do not specify what is the experimental relevance of the protein structures used. Why would binding at the three pockets influence function of the protein?

Happy to clarify this. We do not study how the potential binding pockets influence the function of the protein, but we aim to narrow down the search grid for the subsequent docking simulations to obtain higher throughput. These binding sites are the most probable locations where a newly designed drug (ligand) will bind. Whether or not the binding changes the function of the protein cannot be predicted by AI-Bind. For binding sites with no known functions, further experimental tests are required to fully understand how each binding site influences the function of the protein.

9) Binding probability profiles have very similar shapes. Please discuss the effect of the ligand on the binding site more clearly.

We thank the Reviewer for raising this important question. We now added SI Section 9 and Figure S15, where we address it by showing heatmaps capturing the similarities between the binding probability profiles for different ligands. We find that multiple ligands binding to the same pocket are clustered together in the clustermap. Thus, the binding probability profiles generated by AI-Bind carry information about the molecular structures of the ligands required to potentially bind to each site.

10) Can the method help picking a putative binding site?

The answer is yes, and we are exploring the AI-Bind predicted binding sites in accelerating docking simulations in a future manuscript. We discuss this aspect in the SI Section 13. Currently, we determine the binding sites from the valleys of AI-Bind's binding probability profile by fitting a sine curve to the valleys and considering the region between the points of inflection of the sine curve as the AI-Bind predicted binding site. We compare the AI-Bind predicted pockets to the gold standard experimental data and observe that the AI-Bind predicted binding sites cover 64.05% of all pockets over the 195 different proteins in the gold standard dataset [1].

Minor comments:

11) Annotation of figure 2 should be improved (e.g. meaning of red crosses etc.)

We have updated the caption for Figure 2 accordingly.

12) How is the stereochemistry handled when coming from a 2D representation of the AI engine to the 3D step?

AI-Bind does not take into account the stereochemistry of the molecules, but it uses as input the SMILES of the ligands, and the amino acid sequences of the proteins. AI-Bind uses Mol2vec and ProtVec as the embedding algorithms for the ligands and the proteins, respectively. Both algorithms by default consider the anisomeric structures (2D) of the molecules.

In the validation stage, docking uses 3D structures for both the ligands and the proteins. We downloaded the SDF files for the ligands from PubChem, and the PDB files for the proteins from PDB. The identifiers used for the ligands while searching on PubChem are the InChIKeys. PDB files were obtained from the Protein Data Bank (PDB) by searching with the gene symbols or the UniProt identifiers. The default stereochemistry from the respective databases is used for the docking simulations. Proteins are considered as rigid structures, and we consider rotatable bonds for the ligand molecules.

13) Since docking of macrocycles is challenging due to the conformational restrictions, I suggest giving more details of the procedure in the SI.

We thank the Reviewer for this suggestion. We have now added a detailed description of the docking steps in the Methods section (Lines 477-491). The protein structures used in docking are rigid, whereas we use flexible structures for the ligands (with rotatable bonds).

Overall, the manuscript contains some interesting approaches and studies. The largely in-silico based validation for shown ligands still casts several questions to be clarified and improved upon.

In summary, we wish to thank the Reviewer for prompting us to perform more extensive validation of the binding sites predicted by AI-Bind to the paper, which has undoubtedly improved the quality of our manuscript, and for the many constructive observations on the manuscript.

Reviewer #2 (Remarks to the Author):

AI-Bind: improving binding predictions for novel protein targets and ligands

Chatterjee et al. provided a drug-target prediction model called AI-Bind to generalize for unseen cases. Data imbalance in the training data was solved by supplementing network-derived negative samples. The features for the drugs and proteins were pre-trained from common chemical libraries.

We thank the Referee for the accurate summary of the manuscript. As we discuss next, we modified the main text and supplementary material to address each of the valuable suggestions of the Reviewer.

Major comments:

1) How accurate of the network-derived negative samples are? Many drug-target databases contain only the results of positive samples, i.e. if a drug-target pair is recorded then it is a positive sample, but if a drug-target pair is not recorded then we don't know if it is truly negative, as it may be so that this drug-target pair is never tested.

We thank the Reviewer for prompting us to clarify our negative sampling method. In the manuscript, we provide two major pieces of evidence as to why distant protein-ligand pairs in the Drug-Target Interaction (DTI) network can be used as negative or non-binding pairs. First, we collected experimentally confirmed negative pairs from BindingDB, finding that non-binding occurs at an inhibitor constant (K_i) of 10^6 nM or greater (higher K_i suggests even weaker binding). In Figure 5D, we show how K_i changes as a function of the hop distance within the DTI network. For protein-ligand pairs that are more than 7 hops apart the average K_i is above the non-binding threshold of BindingDB negative pairs, meaning they are most likely non-binding. In Figure 5E, we give an example of a protein-ligand pair (Protein: Serine Racemase, and ligand: Tricarballic acid) which are 7-hops apart in the DTI network and have $K_i = 1.3 \times 10^6$ nM (above the 10^6 nM non-binding threshold set by BindingDB), allowing us to confirm as a true non-binding pair. Similar results hold for the disassociation constant (K_d), and other kinetic constants capturing the strength of binding in concentration units.

Second, in Section S5 we justify why a threshold of 7-hops is a good choice for selecting the network-derived negative samples using EigenSpokes analysis. The protein-ligand pairs which are more than 7-hops apart are far enough from the positive samples and retain significant variability to optimize the training of the machine learning model to learn binding patterns.

2) Data that is used for pre-training of AI-Bind is unclear. The source of compound information is lacking, also, why there are more than half a million proteins?

Since AI-Bind VecNet uses unsupervised pre-training for both protein and small molecule embeddings, we leveraged pre-trained Mol2vec [8] and ProtVec [9] to embed the ligands and the proteins, respectively. Mol2vec is trained on 19.9 M chemicals obtained from ZINC [10] and ChEMBL [11] databases. ProtVec uses as corpus the amino acid sequences of 546,790 proteins

obtained from Swiss-Prot [12]. We have updated Section ‘AI-Bind and statistics across models’ of the Results (Lines 209-211) to clarify this information.

3) Whether the same kind of annotation imbalance on the compounds and targets exist? For example, there exist major chemical structures or protein families that are enriched in the pre-training data that in the end also drive the prediction?

This is an important question related to unsupervised pre-training in machine learning. Large datasets used in pre-training help with the convergence of VecNet to a minimum of the loss function, preferably the global minimum or a minimum suitable for inductive tests [13]. The size of the pre-training corpus is important, and a sufficiently large corpus that includes significantly more structures compared to the training DTI helps improve the inductive performance. There is almost certainly bias in the pre-training data, but the size of the dataset and the low-level nature of the pre-training task, predicting nearby data in sequences, helps fight overfitting. Hence, the bias in the pre-training data will probably lead to better or worse predictions for certain types of ligands or proteins. However, the bias is not likely to over or under-estimate binding.

4) Anidulafungin and cyclosporine have IC50 of 4.64 μ M and 5.82 μ M which are not usually considered as very low concentrations. Therefore, the significance of the experimental validation is limited.

Happy to clarify this. IC50 is not the dosage of the drug but is the amount of a drug needed to inhibit a biological process by 50%, a measure of the therapeutic effect. Therefore, the lower the amount of drug needed, the higher the efficacy of the drug at inhibiting the biological process. This means that a patient will need less concentration of the drug to get the same effect of another drug with a higher IC50, avoiding potential toxic effects that come with higher concentrations. Less than 10 μ M is considered to be a safe IC50 concentration for drugs [14].

5) Only the DeepPurpose was selected for comparison. More systematic comparisons involving other state-of-the-art methods are lacking.

We fully agree with the Reviewer on the need for more systematic comparison with other state-of-the-art methods. Inspired by the Reviewer’s suggestion, we added Lines 226-230 to Section ‘AI-Bind and statistics across models’ and SI Section 8, where we investigated the comparison with another state-of-the-art model, MolTrans [15]. Transductive, semi-inductive, and inductive performances have been summarized in Table S4. AI-Bind achieves the best performances in inductive tests on the network-derived negatives in terms of both AUROC and AUPRC in a 5-fold cross-validation. In this setting, MolTrans achieves an AUROC of 0.619 (\pm 0.021) and an AUPRC of 0.480 (\pm 0.028), while DeepPurpose achieves an AUROC of 0.642 (\pm 0.025) and an AUPRC of 0.583 (\pm 0.016). AI-Bind, on the other hand, achieves an AUROC of 0.742 (\pm 0.029) and an AUPRC of 0.724 (\pm 0.037). These observations validate how unsupervised pre-training of the molecular embeddings helps improve the generalizability of protein-ligand binding prediction models.

In summary, we wish to thank the Reviewer for prompting us to clarify many aspects of our work, which has undoubtedly improved the quality of our manuscript, and for inspiring us to compare AI-Bind with another state-of-the-art model beyond DeepPurpose, broadening the validation and the potential impact of our manuscript.

Reviewer #3 (Remarks to the Author):

In this work, the authors proposed a pipeline, AI-Bind, for predicting protein-ligand interactions with the generalization prediction power of unseen proteins and ligands. The network-based negative sampling, and in-depth analysis of the work could be highlighted. However, some issues must be addressed.

We thank the Referee for highlighting our network-based negative sampling strategy. In the following, we address all the questions and recommendations offered by the Reviewer.

[Major comments]

1) In Introduction, many recent DTI prediction models are neglected in this study. Please cite recent SOTA deep-learning-based models that predict DTIs or DTAs. e.g., MolTrans [1], MONN [2], KGE NFM [3], TransDTI [4], HoTS [5], DTIHNC [6], etc.

*[1] Huang, K., Xiao, C., Glass, L. M. & Sun, J. MolTrans: Molecular interaction transformer for drug target interaction prediction. Bioinformatics btaa880 (2020)
doi:10.1093/bioinformatics/btaa880.*

[2] Li, S. et al. MONN: A Multi-objective Neural Network for Predicting Compound-Protein Interactions and Affinities. Cell Syst 10, 308-322.e11 (2020).

[3] Ye, Q. et al. A unified drug-target interaction prediction framework based on knowledge graph and recommendation system. Nat Commun 12, 6775 (2021).

[4] Kalakoti, Y., Yadav, S. & Sundar, D. TransDTI: Transformer-Based Language Models for Estimating DTIs and Building a Drug Recommendation Workflow. Acs Omega 7, 2706–2717 (2022).

[5] Lee, I. & Nam, H. Sequence-based prediction of protein binding regions and drug-target interactions. J Cheminformatics 14, 5 (2022).

[6] Jiang, L. et al. Identifying drug-target interactions via heterogeneous graph attention networks combined with cross-modal similarities. Brief Bioinform 23, (2022).

Also, on page 3, the authors stated that the SOTA deep learning model ignores the characteristics of proteins and compounds and relies on DTI annotations. However, among recent SOTA studies, there are approaches (e.g., MolTrans, MONN, and HoTS) that learn general features of drugs and proteins in highly sophisticated ways without simply being biased toward protein-ligand and network properties. The expression of the limitations of previous studies should be rephrased.

We thank the Reviewer for pointing us to these relevant references. We now cite them in the Introduction (Lines 63-67). We have also updated the Section ‘Limitations of existing ML models’ in the Results (Lines 178-187), adding the transductive, semi-inductive, and inductive test performances for MolTrans [15]. It is true that the mentioned SOTA models learn beyond the DTI used for training, leveraging the molecular structures to make protein-ligand binding predictions. However, AI-Bind introduces unsupervised pre-training on top of the innovative molecular representations explored in the SOTA models, which helps AI-Bind in generalizing to protein and ligand structures that are different from the ones present in the training data. The comparison between AI-Bind, MolTrans, and DeepPurpose has been summarized in SI Section 8 and Table S4.

2) *The authors mainly reported network characteristics using DTI information collected from BindingDB (Table 1), the same reproduction experimental conditions as the DeepPurpose model (Figure 1). The authors argued that the DTI network has a fat-tailed distribution and said this is a common feature of the DTI network. However, since BindingDB itself is a databank for a collection of experimental assays that particular labs or researchers are interested in, it would not be surprising to observe a small number of hub nodes (proteins or ligands with high connectivity) in this DB.*

The Reviewer is correct and we fully agree. The fat-tailed nature of the degree distribution in the binding datasets is the result of the experimental biases associated with choosing what protein-ligand binding to explore. Some proteins and ligands are indeed studied more than others, hence appearing as hubs in such datasets. Moreover, drugs are often made by small modifications of the same base structure, and thus bind to the same target protein. We consistently observe a fat-tailed annotation behavior in BindingDB, DrugBank, Drug Target Commons, and other databases whose content is driven by preferential attachment mechanism [16]. The proteins which are associated with many biological processes and diseases are studied more often and constitute the hubs in the DTI. We observe fat-tailed annotation distributions for both proteins and ligands in BindingDB (see Figure 1A) and DrugBank (see Figure S7A).

In this manuscript, we are not suggesting that the observed fat-tailed nature of the degree distribution is the discovery of a true property of the DTI network, but rather a feature of the available training databases that must be taken into account to deliver optimal inductive predictions.

3) *If the authors want to report the generalized DTI network properties, the Reviewer suggests collecting more high confident DTI data from additional DBs (e.g., DrugBank, KEGG, IUPHAR, etc.) and re-analyze the network connectivity properties.*

This is an excellent suggestion. In AI-Bind training data, we collected protein-ligand binding pairs from multiple databases like DrugBank, BindingDB, and Drug Target Commons. We observe fat-tailed annotation distributions for both proteins and ligands in the BindingDB training data used in DeepPurpose (Figure 1A). Prompted by the Reviewer's suggestion, we have now separately analyzed the annotation distributions for proteins and ligands in DrugBank, finding a similar fat-tailed behavior. We report these novel observations in SI Section 4 and Figure S7A.

Performance comparisons:

4) *Performance comparisons with other SOTA studies that learn the general features of drugs and proteins should be addressed. Especially, as the authors highlight the generalization power for unseen proteins and unseen ligands in this study, the prediction performance for unseen proteins and unseen ligands should be compared with the aforementioned SOTAs.*

We fully agree with the Reviewer on the need to extend performance comparisons to other SOTA. Initially, we compared AI-Bind to DeepPurpose, and used the performance in inductive tests as a measure of generalizability. However, beyond DeepPurpose, multiple binding

prediction models exist, which use innovative techniques for improved learning of the molecular structures, as pointed out by the Reviewer. For example, the Molecular Interaction Transformer (MolTrans) [15] is a state-of-the-art protein-ligand binding prediction model which uses a combination of sub-structural pattern mining algorithm, interaction modeling module, and an augmented transformer encoder to better learn the molecular structures. Prompted by the Reviewer’s suggestion, we have now added a comparison with MolTrans to the Results (‘AI-Bind and statistics across models’, Lines 226-230) and to SI Section 8. Transductive, semi-inductive, and inductive performances have been summarized in Table S4. AI-Bind performs better in inductive tests in terms of both AUROC and AUPRC. MolTrans achieves an AUROC of 0.619 (± 0.021) and an AUPRC of 0.480 (± 0.028) on unseen proteins and ligands. DeepPurpose achieves an AUROC of 0.642 (± 0.025) and an AUPRC of 0.583 (± 0.016) in the inductive tests. AI-Bind, on the other hand, achieves an AUROC of 0.742 (± 0.029) and an AUPRC of 0.724 (± 0.037) in inductive tests. These observations confirm that unsupervised pre-training of the molecular embeddings helps improve the generalizability of protein-ligand binding prediction models.

5) Also, in Figure 4, besides the 5-fold cross-validation performance, please provide the prediction performance using unseen split data so that readers can confirm the generalized performance.

Thank you — In Figure 4C, we now provide the inductive test performance of VecNet on both unseen proteins and ligands in a 5-fold cross-validation set-up. This summarizes the performance of AI-Bind on unseen split data. The performances are also summarized in Table S3. AI-Bind’s VecNet achieves an AUROC of 0.742 (± 0.029) and an AUPRC of 0.724 (± 0.037) in inductive tests. We provide the error-bars along with the mean performance to show the stability of the performance on the unseen proteins and ligands across different folds.

6) Please compare the results with randomly selected negative samples as well to show the effectiveness of the network-based negative sampling.

We fully agree with the Referee on the importance of a comparison with randomly selected negatives, often used in graph machine learning for link prediction training. Indeed, comparing random negatives with AI-Bind’s network-derived negatives would convey how the network-derived negatives improve the link prediction task. Prompted by the Reviewer’s suggestion, we now discuss this comparison in SI Section 12. In particular, we generate 15 random negative pairs for each binding pair in the AI-Bind training dataset, finding that these balanced random negatives show lower inductive performance compared to the network-derived negatives. Overall, we observe lower inductive test performance on the random negatives (AUROC of 0.709 ± 0.011 and AUPRC of 0.566 ± 0.013) compared to the network-derived negatives (AUROC of 0.745 ± 0.032 and AUPRC of 0.729 ± 0.038). In SI Section 12 we also discuss why random negatives are less informative compared to network-derived negatives in terms of protein-ligand binding.

7) Please provide the result of an ablation study (e.g., w/o network sampling, w/o unsupervised pre-training, etc.) so that the impact of each component of the pipeline can be distinguished.

We thank the Reviewer for this excellent recommendation. In Figure 4C we now address how the performance on unseen proteins and ligands (inductive performance) improves as we sequentially introduce network-derived negatives and unsupervised pre-training in the pipeline (i.e., a typical ablation study set-up). DeepPurpose on the original data includes neither network-derived negatives nor unsupervised pre-training. In this setting, DeepPurpose achieves an AUROC of 0.60 ± 0.066 and an AUPRC of 0.42 ± 0.063 on unseen proteins and ligands, remarkably similar to the proposed duplex configuration model which achieves an AUROC of 0.50 and an AUPRC of 0.30 ± 0.034 in inductive tests. When we introduce network-derived negatives in DeepPurpose training, the inductive performance improves, and we observe an AUROC of 0.642 ± 0.025 and an AUPRC of 0.583 ± 0.016 . Here, it is worth mentioning that DeepPurpose uses end-to-end training, and does not leverage the benefits of unsupervised pre-training. Lastly, we introduce unsupervised pre-training on top of network derived negatives in VecNet, which achieves the best performance on unseen proteins and ligands (AUROC of 0.742 \pm 0.029 and an AUPRC of 0.724 ± 0.037).

AutoDock simulation:

8) In-depth analysis with docking simulation is interesting and could show the robustness of the method. When performing the AutoDock docking simulation, how did the authors determine the initial search box of each target protein for compounds? Need to clarify the steps in the manuscript.

We thank the Reviewer for prompting us to clarify this relevant aspect of our pipeline. We have now added a step-by-step description regarding the docking simulations to the Methods (Lines 477-491).

9) Please provide a statistical difference between the two distributions (e.g., p-value) in Figure 6A.

Prompted by the Referee's suggestion, we performed a Kruskal-Wallis H-test [17] on the binding affinity distributions. We observe that the median binding affinities of the top and the bottom predictions are significantly different, with H statistic equal to 17.76 and a p-value of 2.5×10^{-5} . These observations have been added to the subsection 'Validation of AI-Bind predictions on COVID-19 proteins' at Lines 257-258.

[Minor comments]

10) On page 5, the authors argue that the topology of the protein-ligand interaction network drives the DTI prediction as some deep learning models and a network configuration model demonstrate similar prediction results. Here, the similar prediction performance does not necessarily mean that the models mainly rely on similar features. The expression should be revised.

The Reviewer is correct in stating that similar performances do not imply similar prediction mechanisms. Therefore, we include additional observations in the subsection 'Limitations of

existing ML models' that prompted our conclusion (Lines 149-151). In particular, we claimed that a machine learning model achieving performances comparable to a configuration model (leveraging only degree information) is likely using similar information from the training data, circumventing the encoded molecular structures. To confirm our intuition, we ran an experiment on DeepPurpose described in the main (Table 3), where we randomly shuffled the input amino acid sequences and SMILES, and did not observe any degradation of the transductive test performance. Since random shuffling of the input removes all molecular information, this observation confirms that DeepPurpose does not learn from the molecular structures, rather it uses the degree information from the training DTI. Furthermore, in SI Section 7 and Table S3, we observe that VecNet, the best performing model in inductive tests, is unable to perform well when we replace the molecular embeddings with random numbers (drawn from a uniform random distribution), achieving very similar performance to the configuration model. This means that informative molecular embeddings are required when making inductive binding predictions with VecNet.

In summary, we wish to thank for the many constructive comments offered by the Reviewer. We are impressed by the depth of attention and understanding the Reviewer has offered.

References:

- [1] Cheng, T., Li, X., Li, Y., Liu, Z. & Wang, R. Comparative assessment of scoring functions on a diverse test set. *Journal of Chemical Information and Modeling* 49, 1079–1093 (2009). URL <https://doi.org/10.1021/ci9000053>.
- [2] Krivák, R. & Hoksza, D. P2rank: machine learning based tool for rapid and accurate prediction of ligand binding sites from protein structure. *Journal of Cheminformatics* 10 (2018). URL <https://doi.org/10.1186/s13321-018-0285-8>.
- [3] Lexa KW, Carlson HA. Protein flexibility in docking and surface mapping. *Q Rev Biophys.* 2012 Aug;45(3):301-43. doi: 10.1017/S0033583512000066. Epub 2012 May 9. PMID: 22569329; PMCID: PMC4272345.
- [4] Totrov, M., Abagyan, R. Flexible ligand docking to multiple receptor conformations: A practical alternative. *Current Opinion in Structural Biology* 18, 178-184 (2008). URL <https://doi.org/10.1016/j.sbi.2008.01.004>.
- [5] Kutchukian, P. S., Yang, J. S., Verdine, G. L. & Shakhnovich, E. I. All-atom model for stabilization of α -helical structure in peptides by hydrocarbon staples. *Journal of the American Chemical Society* 131, 4622–4627 (2009). URL <https://doi.org/10.1021/ja805037p>.
- [6] Fujiwara, K., Toda, H. & Ikeguchi, M. Dependence of alpha-helical and beta-sheet amino acid propensities on the overall protein fold type. *BMC Structural Biology* 12, 18 (2012). URL <https://doi.org/10.1186/1472-6807-12-18>.
- [7] Cheng, P.-N., Pham, J. D. & Nowick, J. S. The supramolecular chemistry of β -sheets. *Journal of the American Chemical Society* 135, 5477–5492 (2013). URL <https://doi.org/10.1021/ja3088407>.
- [8] Jaeger, S., Fulle, S. & Turk, S. Mol2vec: Unsupervised machine learning approach with chemical intuition. *Journal of Chemical Information and Modeling* 58, 27–35 (2018). URL <https://doi.org/10.1021/acs.jcim.7b00616>.
- [9] Asgari, E. & Mofrad, M. R. K. Continuous distributed representation of biological sequences for deep proteomics and genomics. *PLOS ONE* 10, e0141287 (2015). URL <https://doi.org/10.1371/journal.pone.0141287>.
- [10] Irwin, J. J.; Shoichet, B. K. ZINC – A Free Database of Commercially Available Compounds for Virtual Screening. *J. Chem. Inf. Model.* 2005, 45 (1), 177– 182, DOI: 10.1021/ci049714+
- [11] Gaulton A, Bellis LJ, Bento AP, Chambers J, Davies M, Hersey A, Light Y, McGlinchey S, Michalovich D, Al-Lazikani B, Overington JP. ChEMBL: a large-scale bioactivity database for

drug discovery. *Nucleic Acids Res.* 2012 Jan;40(Database issue):D1100-7. doi: 10.1093/nar/gkr777. Epub 2011 Sep 23. PMID: 21948594; PMCID: PMC3245175.

[12] Amos Bairoch, Rolf Apweiler, The SWISS-PROT Protein Sequence Data Bank and Its New Supplement TREMBL, *Nucleic Acids Research*, Volume 24, Issue 1, 1 January 1996, Pages 21–25, <https://doi.org/10.1093/nar/24.1.21>

[13] Erhan, D., Courville, A., Bengio, Y., & Vincent, P. (2010). Why Does Unsupervised Pre-training Help Deep Learning? *Proceedings of the Thirteenth International Conference on Artificial Intelligence and Statistics in Proceedings of Machine Learning Research* 9:201-208 Available from <https://proceedings.mlr.press/v9/erhan10a.html>.

[14] Hughes, J., Rees, S., Kalindjian, S. & Philpott, K. Principles of early drug discovery. *British Journal of Pharmacology* 162, 1239–1249 (2011). URL <https://doi.org/10.1111/j.1476-5381.2010.01127.x>.

[15] Huang, K., Xiao, C., Glass, L. M. & Sun, J. MolTrans: Molecular interaction transformer for drug target interaction prediction. *Bioinformatics* btaa880 (2020) doi:10.1093/bioinformatics/btaa880.

[16] Hase T, Tanaka H, Suzuki Y, Nakagawa S, Kitano H. Structure of protein interaction networks and their implications on drug design. *PLoS Comput Biol.* 2009 Oct;5(10):e1000550. doi: 10.1371/journal.pcbi.1000550. Epub 2009 Oct 30. PMID: 19876376; PMCID: PMC2760708.

[17] Kruskal, W. H., & Wallis, W. A. (1952). Use of Ranks in One-Criterion Variance Analysis. In *Journal of the American Statistical Association* (Vol. 47, Issue 260, pp. 583–621). Informa UK Limited. <https://doi.org/10.1080/01621459.1952.10483441>

REVIEWERS' COMMENTS

Reviewer #1 (Remarks to the Author):

The manuscript AI-bind describes a ML-based pipeline for prediction of drug-target interaction. The approach uses a combination of different AI technologies. The predictions are validated extensively. In the revision the authors have performed additional validation studies and added further descriptions. The author's comments also clarified questions.

Overall, the quality of the manuscript has further improved and publication of the research is certainly justified.

Reviewer #2 (Remarks to the Author):

Thanks for the responses. However, a few comments remain:

1. Figure 2A-B the right panel: it is unclear of the interpretation. It seems that increasing degrees are associated with decreasing degree ratios. Does it suggest that a hub node tends to have lower ratio of positive associations? If so then it is contradictory to the texts in Figure 2 legend, 'the hubs get many positive or binding annotations, whereas the low degree nodes get both binding and non-binding annotations.'
2. Figure 2C: top 100 false positives and false negatives were defined for proteins. However, the prediction was made for a particular protein-ligand pair, not for a protein. Therefore, it is unclear under what criteria a protein is called a false positive (or negative).
3. Equation (6): it is unclear what is $P^{(0,0)}$.
4. Equation (9): does it suggest that the prediction of unseen protein-ligand pairs is the same, as $L^{(1,0)}$ and $L^{(0,1)}$ are constant?
5. Prediction of the top/bottom ligands for COVID-19 should be listed as supplementary tables.
6. Data preparation method: what does 'filter out all samples outside the temperature range' mean?

Reviewer #3 (Remarks to the Author):

The reviewer's concerns were adequately addressed by the authors, and the manuscript could be published in its present form.

Reviewer #1 (Remarks to the Author):

The manuscript AI-bind describes a ML-based pipeline for prediction of drug-target interaction. The approach uses a combination of different AI technologies. The predictions are validated extensively. In the revision the authors have performed additional validation studies and added further descriptions. The author's comments also clarified questions.

Overall, the quality of the manuscript has further improved and publication of the research is certainly justified.

We wish to thank the Reviewer for the multiple constructive observations and in particular, for prompting us to provide further details on the docking-based validation and for suggesting further experiments to validate the AI-Bind predicted binding sites. These additions have undoubtedly improved the quality of the manuscript.

Reviewer #2 (Remarks to the Author):

Thanks for the responses. However, a few comments remain:

We thank the Reviewer for the further constructive comments. In the following, we address the remaining recommendations of the Reviewer.

1. Figure 2A-B the right panel: it is unclear of the interpretation. It seems that increasing degrees are associated with decreasing degree ratios. Does it suggest that a hub node tends to have lower ratio of positive associations? If so then it is contradictory to the texts in Figure 2 legend, 'the hubs get many positive or binding annotations, whereas the low degree nodes get both binding and non-binding annotations.'

We thank the Reviewer for prompting us to clarify this very important aspect of our work. The former Figures 2a-b (right panels) do not suggest that a hub node tends to have a lower ratio of positive annotations. Indeed, on average, the hub nodes exhibit many positive annotations as the kinetic constants $\langle K_d \rangle$ associated with the hubs are lower compared to other nodes with smaller degrees (former Figures 2a-b, middle panels). Since the variability in $\langle K_d \rangle$ is large for the low-degree nodes, they tend to have both positive and negative annotations. Many low-degree nodes have degree ratios close to 0, suggesting that most or all the associated annotations are non-binding. On the other hand, the higher degree nodes tend to have more positive annotations, and hence nonzero degree ratios. Under a random train-validation-test split of the edges in DeepPurpose, the hubs contribute to the majority of the samples in both training and testing, and the monotonous relation between k and $\langle K_d \rangle$ helps DeepPurpose in making an accurate prediction for the hubs leveraging only the degree information.

We agree with the Reviewer that the scatter plots in former Figures 2a-b (right panels) were misleading due to data point density and can create the impression of a negative correlation between degree and degree ratio. Prompted by the Reviewer's comment, we have modified Figure 2a-b (now Figure 1b-c) and expanded SI Section 1 with further details regarding the relation between node degrees and kinetic constants.

2. Figure 2C: top 100 false positives and false negatives were defined for proteins. However, the prediction was made for a particular protein-ligand pair, not for a protein. Therefore, it is unclear under what criteria a protein is called a false positive (or negative).

We are delighted to clarify this. False positives and false negatives are associated with the protein-ligand pairs. In Figure 2c, a false positive corresponds to a protein-ligand pair which is predicted by DeepPurpose to be binding, but reported as non-binding in BindingDB. On the other hand, a false negative pair is predicted to be non-binding by DeepPurpose, but reported in BindingDB as binding. We show in Figure 2c that these false predictions are highly correlated with the degree ratio of the protein in the protein-ligand pair under consideration. If a non-binding pair (from BindingDB) contains a protein with large degree ratio, DeepPurpose predicts the pair as binding and makes a false positive prediction. The opposite is observed for the false negative pairs, which are associated with the proteins with low degree ratios. Prompted by the Reviewer's comment, we now clarify in Figure 2c's caption that the false positives and the false negatives correspond to protein-ligand pairs and not individual proteins.

3. Equation (6): it is unclear what is $P^{(0,0)}$.

In the Methods subsection “Network Configuration Model – Overview”, we describe three types of links in protein-ligand interactions: $m=(1,0)$ represents the positive or binding interactions, $m=(0,1)$ refers to the negative or non-binding interactions, and $m=(0,0)$ represents the absence of any annotation, i.e., the protein-ligand pairs for which the binding information is unknown. In Eqs. (5) and (6), we analytically derive the expressions for $p_{ij}^{(1,0)}$ and $p_{ij}^{(0,1)}$, respectively. These equations represent the probabilities of positive and negative links predicted by the duplex configuration model. We further mention the constraint: $p_{ij}^{(1,0)} + p_{ij}^{(0,1)} + p_{ij}^{(0,0)} = 1$. Since no protein-ligand pair can have both positive and negative links in reality, the probability of the absence of annotations predicted by the configuration model is $p_{ij}^{(0,0)} = 1 - p_{ij}^{(1,0)} - p_{ij}^{(0,1)}$.

4. Equation (9): does it suggest that the prediction of unseen protein-ligand pairs is the same, as $L^{(1,0)}$ and $L^{(0,1)}$ are constant?

The Reviewer is correct. In an inductive scenario, we have no degree information for the unseen protein and the unseen ligand. Consequently, for the configuration model purely based on degree information, the conditional probability $p_{i,j}^{*conditional}$ is a constant function of the observed average values of positive and negative link probabilities, which are related to $L^{(1,0)}$ and $L^{(0,1)}$.

5. Prediction of the top/bottom ligands for COVID-19 should be listed as supplementary tables.

We appreciate the suggestion. The top and bottom predictions are publicly shared via our open-source GitHub page. We now share also the following supplementary tables:

Top predictions validated by docking: top_predictions_validated_docking.xlsx

Top 100 predictions made by AI-Bind: top_100_predictions.xlsx

Bottom 100 predictions made by AI-Bind: bottom_100_predictions.xlsx

6. Data preparation method: what does ‘filter out all samples outside the temperature range’ mean?

Thank you for giving us the chance to further elaborate on this. Multiple protein-ligand pairs in BindingDB report the temperature associated with the binding reactions along with other information like the kinetic constants. While preparing the positive samples for the AI-Bind training data, we filtered out the pairs for which the binding reaction took place in $<20^{\circ}\text{C}$ or $>45^{\circ}\text{C}$. At temperatures above 45°C the proteins are more likely denatured (unraveling of the structure), while below 20°C the cold temperatures also denature the proteins as well by altering the tertiary structure to a form more stable in the colder temperatures. By filtering out these extremes, we aim to select data consistent with functional proteins within the human body.

In summary, we wish to thank the Referee for prompting us to clarify and reformulate key aspects of our work.

Reviewer #3 (Remarks to the Author):

The reviewer's concerns were adequately addressed by the authors, and the manuscript could be published in its present form.

We thank the Reviewer for all the constructive suggestions, especially comparing AI-Bind with another state-of-the-art binding prediction model, which has significantly improved the confidence on AI-Bind as a tool in drug discovery. Incorporating all the suggestions has notably improved the quality of the manuscript.